
SciPost Phys. Lect Notes 22 (2021)

# Lectures on bulk reconstruction

**Nirmalya Kajuri**$^\star$

Chennai Mathematical Institute, H1 Sipcot IT Park, Tamil Nadu, India

$\star$ nkajuri@cmi.ac.in

## Abstract

If the AdS/CFT conjecture holds, every question about bulk physics can be answered by the boundary CFT. But we still don't know how to translate all the questions about bulk physics to questions about the boundary CFT. Completing this bulk-boundary dictionary is the aim of the bulk reconstruction program, which we review in these lectures. We cover the HKLL construction, bulk reconstruction in AdS/Rindler, mirror operator construction of Papadodimas and Raju, and the Marolf-Wall paradox. These notes are based on lectures given at ST4 2018.

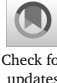
## Contents

According to the AdS/CFT conjecture, any conformal field theory in a $d$-dimensional space-time is equivalent to a theory of quantum gravity in a $d+1$-dimensional spacetime. Therefore one should be able to use it to try to learn about bulk physics from the boundary.

If the bulk and boundary theories are indeed equivalent, any question one may ask about bulk physics can in principle be answered by the boundary CFT. The CFT in the boundary *is* the bulk quantum theory of gravity!

In principle, the boundary theory knows the answers to all the outstanding questions like:
1. What happens when an observer crosses the black hole horizon? Do they see a smooth horizon or a firewall? [1]
2. More generally, how is the black hole information loss paradox [2] resolved?
3. What happens at singularities, where classical general relativity breaks
down.

If we could solve the boundary CFT completely we should have answers to all these questions about bulk physics.

So why haven't we solved all of these problems already? The reason is that to answer questions about bulk physics we would need to know how to translate them to questions about the boundary conformal field theory. The bulk and boundary theories are formulated in terms of completely different spaces and operators, and we would need to know how to map all the observables of one theory to the observables of the other. In other words, we would need the complete dictionary between the two theories. The original formulation of AdS/CFT conjecture does give us a dictionary, but it is only a partial dictionary.

The CFT may well have all the answers, but we don't know how to ask all the right questions!

The topic of these lectures is the ongoing program of 'bulk reconstruction', which aims to complete the bulk-boundary dictionary. This program is still at an early stage. We can translate semiclassical observables in the bulk to the boundary CFT. But quantum gravity questions remain well out of reach. Even in the absence of gravity, observables that lie beyond black hole horizons are problematic. In these lectures, we will review the progress in these topics.

The plan of the lectures is as follows. First, we will present an overview of the program. Then we will briefly recall the AdS/CFT dictionary and proceed to make a precise statement of the program. In section 2 we will show how to obtain a boundary representation of a free scalar field in pure AdS by solving the equations of motion. Sections 3 and 4 will deal with interacting scalars. We will cover the case of a self-interacting scalar field as well as those of scalars interacting with gauge or gravitational fields. We will consider reconstruction in AdS/Rindler wedges in section 5, which has certain novel features. A different method of reconstruction via symmetries will be demonstrated in section 6. Section 7 outlines challenges to bulk reconstruction behind a black hole horizon. In this section, we cover the Papadodimas-Raju proposal and the Marolf-Wall paradox. The final section presents our conclusions.

A notable omission from these lectures is the concept of entanglement wedge reconstruction. This is an important topic, but we will only say a few words about it.

An existing review that covers these topics and has influenced our presentation is the TASI lectures by Harlow [3]. Another review that overlaps with this one is [4].

A note about notation. We will mostly use $(r, t, \Omega)$ coordinates for the bulk spacetime. Then

the boundary coordinates are just $(t, \Omega)$. Occasionally when we want to distinguish between the bulk and boundary co-ordinates clearly we will use $y$ to denote bulk coordinates and $X$ for boundary coordinates.

# 1 Overview of the program

In the last section, we spoke about 'completing the dictionary' without precisely defining what it means. Our aim in this section is to make this more precise. To do this, we first review the original AdS/CFT dictionary and learn what it is already telling us about bulk physics. We then outline the broad goals of the program.

## 1.1 The AdS/CFT dictionary

The AdS/CFT correspondence [5–7] is usually stated as the equality of the partition functions of the bulk and boundary theories.

A different formulation of the correspondence, which is expected to be equivalent to the statement above, is the extrapolate dictionary [8–10]. We state the extrapolate dictionary for scalar fields[1]:

$$\lim_{r \to \infty} r^{n\Delta} \langle \phi(r, t_1, \Omega_1) \phi(r, t_2, \Omega_2) ..... \phi(r, t_n, \Omega_n) \rangle_{\text{Pure AdS}}$$
$$= \langle 0| \mathcal{O}(t_1, \Omega_1) \mathcal{O}(t_2, \Omega_2) .... \mathcal{O}(t_n, \Omega_n) |0\rangle. \tag{1}$$

Here $\mathcal{O}$ is the scalar primary dual to the bulk scalar $\phi$. It has dimension $\Delta$ which is related to the mass $M$ of the scalar field as $\Delta = \frac{d}{2} + \frac{1}{2}\sqrt{d^2 + 4M^2}$ where $d$ is the number of space dimensions. A similar dictionary can be written down for other fields.

This was for pure AdS. More generally, for any semi-classical asymptotically AdS geometry $g$ we expect that there will be a dual state $|\psi_g\rangle$

$$g \longleftrightarrow |\psi_g\rangle \tag{2}$$

such that

$$\lim_{r \to \infty} r^{n\Delta} \langle \phi(r, t_1, \Omega_1) \phi(r, t_2, \Omega_2) ..... \phi(r, t_n, \Omega_n) \rangle_g = \langle \psi_g| \mathcal{O}(t_1, \Omega_1) \mathcal{O}(t_2, \Omega_2) .... \mathcal{O}(t_n, \Omega_n) |\psi_g\rangle. \tag{3}$$

(1) is a special case of this where the geometry $g$ is pure AdS and the dual state is the CFT vacuum $|0\rangle$:

Pure AdS $\longleftrightarrow |0\rangle$.

Another example of a semiclassical asymptotically AdS spacetime is the two-sided eternal black hole. The eternal black hole has two asymptotic boundaries. Consequently, the state dual to the eternal black hole must belong to the tensor product of the Hilbert Spaces of the two CFTs on the two boundaries. The correct dual state is the thermofield double state [12]:

$$\text{Eternal Black Hole} \longleftrightarrow \frac{1}{\sqrt{Z(\beta)}} \sum_E e^{-\frac{\beta E}{2}} |E\rangle |E\rangle, \tag{4}$$

---

[1]For interacting scalar fields in pure AdS, this equivalence has been proven [11].

where the sum is over energy eigenstates, $\beta$ is the inverse temperature of the black hole, and $Z(\beta)$ the partition function at inverse temperature $\beta$.

In general, we don't know the bulk dual of a given boundary state.

The extrapolate dictionary already has some information about bulk physics. We can do "scattering experiments" where we send in wave-packets from close to the boundary, have them scatter, and collect them later close to the boundary. The result of such "scattering experiments"[2] will be contained in the CFT correlator $\langle O(X_1)O(X_2)O(X_3)O(X_4)\rangle$.

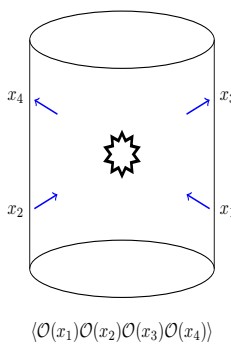

$$\langle \mathcal{O}(x_1)\mathcal{O}(x_2)\mathcal{O}(x_3)\mathcal{O}(x_4)\rangle$$

Figure 1: A bulk 'scattering experiment'.

But this does not cover all bulk information. For instance, if we want to know the correlator between bulk fields for finite values of $r$, which may be useful for the description of a local bulk experiment, that cannot be answered by the extrapolate dictionary directly. One would need to develop the bulk-boundary dictionary further.

## 1.2 Statement of the program

After the general discussion in the last section, we are now ready to state the program we will follow from here on. To do this we first need to specify the regime we will be working in.

As we discussed earlier, we will always work in the regime where bulk geometry is semi-classical. The condition for semiclassicality is that the gravitational constant $G \ll \ell^{d-1}$ where $\ell$ is the AdS radius.

For a CFT to have a semiclassical bulk dual, a set of conditions have to be fulfilled. We refer the reader to [15, 16] for discussions on this issue. One key condition is that the CFT should have a parameter $N \gg 1$ which controls the factorization of the correlators of the primary operators which are dual to bulk fields. This means if the two-point function of such a primary operator is normalized to be of $\mathcal{O}(1)$, then higher correlators are suppressed as

$$\langle \mathcal{O}_i \mathcal{O}_j \mathcal{O}_k \rangle \sim 1/N^a \,,$$

where $a$ is some $\mathcal{O}(1)$ number. In particular, the correlators will follow Wick contraction for infinite $N$. $N$ is dual to the perturbative parameter in the bulk field theory, where a similar factorization takes place in bulk correlators.

---

[2] We put scattering experiment in quotes because, unlike in flat space, defining wave packets in the boundary is problematic in AdS. [13, 14]

In the CFTs known to have a bulk dual, the role of $N$ is played by the central charge of the CFT. $N$ is the expansion parameter in the CFT and it is related to the gravitational constant as

$$N^2 = \frac{\ell^{d-1}}{G}.$$ (5)

We refer to [17] for the stringy origins of the duality and the role of $N$.

Note that $N$ is the only expansion parameter so far in the CFT. The most general dual bulk theory with Einstein gravity and scalar fields will therefore have an action like this (in units where AdS radius is one)[3]:

$$
\begin{aligned}
S = &\frac{1}{G} \int d^{d+1}y \sqrt{-g} R + \int d^{d+1}y \sqrt{-g} \left( \partial_\mu \phi \partial^\mu \phi + M^2 \phi^2 \right) \\
&+ \lambda \sqrt{G} \int d^{d+1}y \sqrt{-g} \left( \frac{\phi^3}{3!} + \text{all possible cubic couplings} \right) \\
&+ \lambda' G \int d^{d+1}y \sqrt{-g} \left( \frac{\phi^4}{4!} + \text{all possible quartic couplings} \right) + \dots\dots\dots
\end{aligned}
$$ (6)

where $\lambda, \lambda'$ are $\mathcal{O}(1)$ numbers. The strengths of the couplings are tightly constrained. A general bulk field theory may have couplings of widely different strengths (eg. standard model), but a theory with a holographic CFT dual can't unless there are more expansion parameters present.[4]

For the class of theories discussed above, the extrapolate dictionary (3) gives us a way to relate bulk fields near the boundary to boundary CFT operators. But it does not tell us how to translate the bulk fields deep inside the bulk to boundary operators.

The goal of the bulk reconstruction program is to discover CFT operators that represent bulk fields at all bulk points. That is, $\phi_{CFT}$ which satisfy:

$$\langle \phi(r_1, t_1, \Omega_1) \phi(r_2, t_2, \Omega_2) \rangle_g = \langle \psi_g | \phi_{CFT}(r_1, t_1, \Omega_1) \phi_{CFT}(r_2, t_2, \Omega_2) | \psi_g \rangle.$$ (7)

In the next chapter, we will see how to find $\phi_{CFT}$.

## 2 Boundary representation of free fields in the bulk

In this section, we will review the techniques for finding CFT representations for free fields in the bulk. First, we briefly review field theory in AdS.

### 2.1 Free scalar fields in AdS

The AdS metric is given by:

$$ds^2 = -\left(1 + \frac{r^2}{\ell^2}\right) dt^2 + \frac{dr^2}{1 + \frac{r^2}{\ell^2}} + r^2 d\Omega_{d-1}^2,$$ (8)

where $\ell$ is the AdS radius.

Henceforth we will put $\ell = 1$.

---

[3]If there are higher derivative terms on the gravity side they will be controlled by the 'tHooft parameter. [17]
[4]We thank Daniel Kabat for a discussion on this point.

As we had discussed earlier, we will work in the semiclassical regime where the bulk action is given by (6). By taking the $G \to 0$ limit($N \to \infty$ limit in the CFT) we get the free field equation in pure AdS:

$$(\Box - M^2)\phi = 0, \tag{9}$$

where $\Box$ is the D'Alembartian in anti-de Sitter spacetime. and $M$ is the mass parameter for the field $\phi$.

In this limit, gravity is switched off. So we can neglect gravity and consider the scalar field in a fixed background. Let us obtain the quantum theory of this field.

From rotational and time translation symmetry of the metric (8) we know that the solution to (9) will be of the form $f_{\omega l \vec{m}}(r, t, \Omega) = \psi_{\omega l}(r) e^{-i\omega t} Y_{l \vec{m}}(\Omega)$ where $Y_{l \vec{m}}(\Omega)$ are the usual spherical harmonics.

Substituting this in (9) gives:

$$(1 + r^2)\psi'' + \left(\frac{d-1}{r}(1 + r^2) + 2r\right)\psi' + \left(\frac{\omega^2}{1 + r^2} - \frac{l(l + d - 2)}{r^2} - M^2\right)\psi = 0. \tag{10}$$

At large $r$ this becomes

$$r^2 \psi'' + (d + 1)r \psi' - M^2 \psi = 0. \tag{11}$$

Clearly, this has polynomial solutions of the form $r^{-\alpha}$. Substituting $\psi(r) = r^{-\alpha}$ in the above gives us two independent solutions, $\alpha = \Delta, d - \Delta$ where:

$$\Delta = \frac{d}{2} + \frac{1}{2}\sqrt{d^2 + 4M^2}. \tag{12}$$

So the asymptotic solution to (9) will have the form:

$$\phi(r, t, \Omega) = r^{\Delta - d} K(t, \Omega) + r^{-\Delta} L(t, \Omega). \tag{13}$$

Normalizable modes are the ones with $r^{-\Delta}$ fall-off. These are the ones we need to define a unitary field theory in AdS. Note that it is the same $\Delta$ that appeared in (3).

We further impose smoothness at $r = 0$ which quantizes $\omega$:

$$\omega_{nl} = \Delta + l + 2n, \tag{14}$$

where $n = 0, 1, 2....$

The full solution for $f_{\omega l \vec{m}}(r, t, \Omega)$ is given by:

$$f_{\omega l \vec{m}}(r, t, \Omega) = \frac{1}{N_{\Delta nl}} e^{-i\omega_{nl} t} Y_{l \vec{m}}(\Omega) \left(\frac{r}{\sqrt{1 + r^2}}\right)^l \left(\frac{1}{\sqrt{1 + r^2}}\right)^\Delta$$
$$\phantom{f_{\omega l \vec{m}}(r, t, \Omega) = } {}_2F_1\left(-n, \Delta + l + n, l + d/2, \frac{r^2}{1 + r^2}\right), \tag{15}$$

where $N_{\Delta nl}$ is the normalization constant.

Now that we have the modes we can quantize the fields.

$$\phi(r, t, \Omega) = \phi^- + \phi^+ = \sum_{nl\vec{m}} f_{\omega l \vec{m}}(r, t, \Omega) a_{\omega l \vec{m}} + f^*_{\omega l \vec{m}}(r, t, \Omega) a^\dagger_{\omega l \vec{m}}, \tag{16}$$

where $a, a^\dagger$ are the annihilation and creation operators. They create normalizable particle excitations in the bulk.

## 2.2 Free field reconstruction in mode sum approach

We want to recreate the free scalar field of the last section as a CFT operator. That is, we want a CFT operator that satisfies:

$$\langle \phi(r_1, t_1, \Omega_1) \phi(r_2, t_2, \Omega_2) \rangle_{\text{Pure AdS}} = \langle 0 | \phi_{CFT}(r_1, t_1, \Omega_1) \phi_{CFT}(r_2, t_2, \Omega_2) | 0 \rangle. \qquad (17)$$

It is enough to consider only two point functions because in free field theory higher-order correlators factorize to products of two point functions. The dual phenomenon in the CFT is large $N$ factorization.

How do we obtain such a $\phi_{CFT}$? This problem was originally solved in [18–20]. The construction below is known as the HKLL construction after Hamilton, Kabat, Lifschytz, and Lowe. They did some of the pioneering work in this field.

To obtain this representation, we first note that the bulk field satisfies the free field equation:

$$(\Box - M^2)\phi = 0. \qquad (18)$$

We also note that the extrapolate dictionary looks like a boundary condition for the bulk field:

$$\lim_{r \to \infty} r^\Delta \phi(r, t, \Omega) = \mathcal{O}(t, \Omega). \qquad (19)$$

This equation relates the boundary value of the field to a primary operator in the conformal field theory. This suggests that if we solve (9) with (19) as the boundary condition we would get an expression for $\phi$ in terms of CFT operators $\mathcal{O}$.

Of course, (19) is not really a boundary condition as it maps fields between two different spaces. The right-hand side is a CFT operator that acts on the CFT Hilbert space and the left-hand side is the boundary value of a bulk field. So what we will really do is to try to find a CFT operator $\phi_{CFT}(r, t, \Omega)$ which satisfies:

$$(\Box - M^2)\phi_{CFT}(r, t, \Omega) = 0. \qquad (20)$$

Here $r$ can be thought of as a parameter which this CFT operator depends on. Then we demand that in the limit where this parameter becomes large, $\phi_{CFT}$ is given by (19). Then we solve for this CFT operator.

This is the right way of thinking about bulk reconstruction, but as far as the logistics of solving the problem is concerned, it is exactly the same as solving (9) as a bulk equation of motion as a boundary value problem with (19) as the boundary value. In the relevant literature this distinction is not made. The bulk field $\phi$ and its CFT representation $\phi_{CFT}$ are usually denoted by the same notation. Now that we know what is going on, we too will drop the distinction and denote $\phi_{CFT}$ as just $\phi$ from here on, except when there is any possibility of confusion between the bulk field and its CFT representation.

Let us now solve the problem. We should note that (19) is not a standard boundary value problem. In field theory, we usually specify initial conditions on a spacelike Cauchy surface. In this case, we are specifying boundary conditions on a timelike surface. This is not a well-studied problem in mathematics. As we will see, the solution will turn out not to be unique.

That said, it is fairly straightforward to solve this boundary value problem in this case. For simplicity, we will consider the case where $\Delta$ is an integer. Then the solution becomes periodic in time and we can limit the range of $t$ to $-\pi$ to $\pi$. For the general case, we refer the reader to [19, 20].

We start from the expansion (16) and plug it in (19):

$$\lim_{r \to \infty} r^\Delta \phi(r, t, \Omega) = \lim_{r \to \infty} r^\Delta \sum_{nl\vec{m}} f_{\omega l \vec{m}}(r, t, \Omega) a_{\omega l \vec{m}} + f^*_{\omega l \vec{m}}(r, t, \Omega) a^\dagger_{\omega l \vec{m}} = \mathcal{O}(t, \Omega). \qquad (21)$$

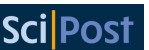

Now

$$
\lim_{r\to\infty} r^{\Delta} f_{\omega l \vec{m}}(r,t,\Omega)
$$

$$
= \lim_{r\to\infty} r^{\Delta} \frac{1}{N_{\Delta nl}} e^{-i\omega_{nl}t} Y_{l\vec{m}}(\Omega) (\frac{r}{\sqrt{1+r^2}})^l (\frac{1}{\sqrt{1+r^2}})^{\Delta} \left( {}_2F_1\left(-n, \Delta+l+n, l+d/2, \frac{r^2}{1+r^2}\right) \right)
$$

$$
= \frac{1}{N_{\Delta nl}} e^{-i\omega_{nl}t} Y_{l\vec{m}}(\Omega) {}_2F_1\left(-n, \Delta+l+n, l+d/2, 1\right) \tag{22}
$$

$$
:= g_{\omega l \vec{m}}(t,\Omega). \tag{23}
$$

Then (21) simplifies to

$$
\sum_{nl\vec{m}} g_{\omega l \vec{m}}(t,\Omega) a_{\omega l \vec{m}} + g^*_{\omega l \vec{m}}(t,\Omega) a^{\dagger}_{\omega l \vec{m}} = \mathcal{O}(t,\Omega). \tag{24}
$$

When $\Delta$ is an integer $g_{\omega lm}$ are orthogonal to all $g^*_{\omega lm}$. We would now like to invert this relation using the orthonormality and completeness of the functions $e^{-i\omega_{nl}t}$ and $Y_{l\vec{m}}(\Omega)$. So we define:

$$
\tilde{g}_{\omega l \vec{m}}(t,\Omega) = \frac{N_{\Delta nl}}{{}_2F_1\left(-n, \Delta+l+n, l+d/2, 1\right)} e^{-i\omega_{nl}t} Y_{l\vec{m}}(\Omega).
$$

Then we can solve for $a$:

$$
a_{\omega l \vec{m}} = \int_{-\pi}^{\pi} dt \int d\Omega \, \tilde{g}^*_{\omega l \vec{m}}(t,\Omega) \mathcal{O}(t,\Omega). \tag{25}
$$

Similarly for $a^{\dagger}_{\omega lm}$.
Plugging it back we get

$$
\phi(r,t,\Omega) = \sum_{nl\vec{m}} f_{\omega l \vec{m}}(r,t,\Omega) \int_{-\pi}^{\pi} dt' \int d\Omega' \, \tilde{g}^*_{\omega l \vec{m}}(t',\Omega') \mathcal{O}(t',\Omega')
$$

$$
+ f^*_{\omega l \vec{m}}(r,t,\Omega) \int_{-\pi}^{\pi} dt'' \int d\Omega'' \, \tilde{g}_{\omega l \vec{m}}(t'',\Omega'') \mathcal{O}(t'',\Omega'')
$$

$$
= \sum_{nl\vec{m}} \int_{-\pi}^{\pi} dt' \int d\Omega' \left( \sum_{nl\vec{m}} f_{\omega l \vec{m}}(r,t,\Omega) \tilde{g}^*_{\omega l \vec{m}}(t',\Omega') + c.c \right) \mathcal{O}(t',\Omega'). \tag{26}
$$

It turns out that $\sum_{nl\vec{m}} f_{\omega l \vec{m}}(r,t,\Omega) \tilde{g}^*_{\omega l \vec{m}}(t',\Omega')$ is real and therefore equal to its complex conjugate. This gives the final form for the expression:

$$
\phi(r,t,\Omega) = \int_{-\pi}^{\pi} dt' \int d\Omega' K(r,t,\Omega;t',\Omega') \mathcal{O}(t',\Omega'), \tag{27}
$$

where

$$
K(r,t,\Omega;t',\Omega') = 2 \left( \sum_{nl\vec{m}} f_{\omega l \vec{m}}(r,t,\Omega) \tilde{g}^*_{\omega l \vec{m}}(t',\Omega') \right). \tag{28}
$$

This is known as the smearing function.
Using (22) we have that

$$
K(r,t,\Omega;t',\Omega') \propto \sum_{nl\vec{m}} f_{\omega l \vec{m}}(r,t,\Omega) e^{-i\omega_{nl}t} Y_{l\vec{m}}(\Omega). \tag{29}
$$

So the smearing function is proportional to the Fourier transform of the mode functions.

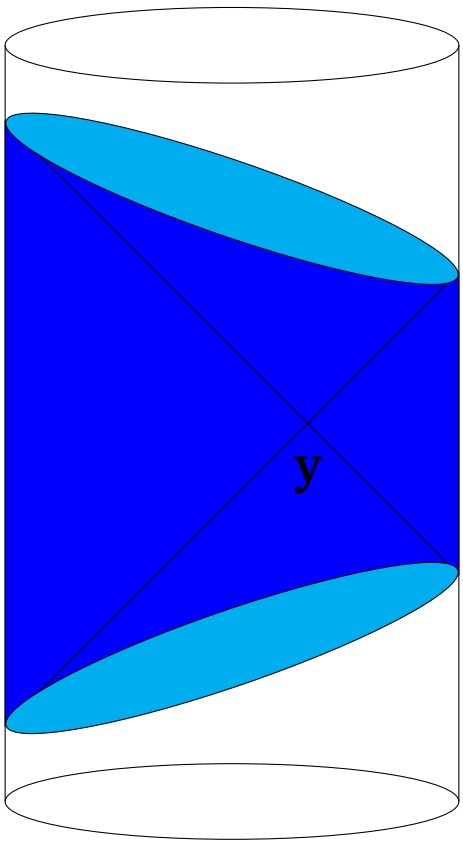

Figure 2: The boundary representation of a bulk scalar field at a point y has support on all boundary points spacelike separated from y.

We note that the smearing function is not unique. We can see from (14) that only modes between $-\Delta$ and $\Delta$ appear in the solution for $\mathcal{O}(t,\Omega)$. Therefore if we add a term $e^{ikt}$ to the smearing function where $k$ is any integer between $-\Delta + 1$ and $\Delta - 1$, the integration $\int dt\, e^{ikt}\mathcal{O}(r,t,\Omega)$ vanishes. So we can add any term of the form $\sum_k c_k e^{ikt}$ to the smearing function without changing (27).

This freedom allows us to put the smearing function in a convenient form. In particular, we can arrange for the smearing function to be non-zero only at boundary points spacelike separated from the bulk point $(r,t,\Omega)$. This is the minimal support that it can have (We refer to section 2.3 of [19] for details).

We now have an expression for the boundary representation of the bulk field. By writing the bulk coordinate as $y$ and boundary coordinate as $X$ we can denote it simply as:

$$\phi(y) = \int dX K(y;X)\mathcal{O}(X). \tag{30}$$

Where the range of integration is over all points $X$ in the boundary which are spacelike separated from the bulk point $y$. Note that this is a non-local operator in the CFT.

Now that we have the CFT representation $\phi(r,t,\Omega)$ we can check whether it indeed satisfies the condition (17). Let us sketch the steps of the check. First we note that

$$\langle 0|\phi(y)\phi(y')|0\rangle = \int dX dX' K(y;X)K(y';X')\langle 0|\mathcal{O}(X)\mathcal{O}(X')|0\rangle. \tag{31}$$

Where we have used (30). Now $\langle 0|\mathcal{O}(X)\mathcal{O}(X')|0\rangle$ is fixed completely by symmetry. We can therefore easily evaluate the above equation. As may be expected, it turns out to give the

correct bulk two point function. The information about the bulk has been encoded in the boundary operator through the smearing function.

Here we worked in global coordinates but we could have worked in the Poincare coordinates. That gives a smearing function with support on the boundary of the Poincare patch. This matches with the global smearing function in the Poincare patch coordinates, up to the ambiguities in the definition of the smearing function mentioned above. We refer to Appendix C of [20] for details.

The generalization to higher-spin fields can be carried out straightforwardly [21–23]. For bulk reconstruction in the background of a BTZ black hole see [24]. An interesting new technique for finding a representation for the bulk field using modular Hamiltonians was given in [25]. This technique can be used to find a CFT representation for the bulk field in a variety of backgrounds.

# 3 Boundary Representation for interacting fields

From the CFT point of view, our program is to find the operator that represents the bulk field at finite $N$. We can try to approximate this in a perturbation series in $1/N$:

$$\phi = \phi^{(0)} + \frac{1}{N}\phi^{(1)} + \frac{1}{N^2}\phi^{(2)} + .. \tag{32}$$

In the last section, we obtained the 0th order approximation. In this section, our aim is to obtain higher-order corrections in $1/N$.

On the bulk side, the perturbation series above translates to a perturbative expansion in $1/\sqrt{G}$. The 0th order approximation in CFT corresponded to the free field equation in the bulk. Obtaining the higher-order corrections in $1/N$ on the CFT side is equivalent to including interactions in the bulk theory. We have to take an interacting field in the bulk, expand it in $1/\sqrt{G}$ and try to obtain the boundary representation.

In this section, we will see how to do that. This can be done in several ways. One of them is an extension of the idea we used in the last section, which is to treat bulk reconstruction as a boundary value problem. One can then introduce an appropriate Green's function and solve the interacting theory order by order. We will discuss this in the next section.

Another approach is to fix the $1/N$ corrections by demanding that they satisfy microcausality (i.e spacelike separated fields should commute) in the bulk. We will discuss this as well.

There are other approaches that give the same result which we have not discussed here. A new and interesting new approach is the one in [26], which obtains the corrections by demanding that the boundary representation of a bulk field behaves like a good CFT operator. This demand fixes the operator at $1/N$ order.

## 3.1 Interacting scalars through the Green's function method

Boundary representation for interacting scalars using the Green's function method was first considered in [27] and generalized to even dimensions in [28]. We outline this method below.

Let us consider an interacting $\phi^3$ theory:

$$(\Box - M^2)\phi = \frac{\lambda}{N}\phi^2, \tag{33}$$

where $\lambda$ is a $\mathcal{O}(1)$ number.

In this section, we will obtain a CFT representation for this bulk field. Once again the strategy is to solve the bulk equation of motion. For an interacting theory, it is useful to do this using Green's function method.

In the last section we saw that we can arrange it so that the smearing function for a bulk point has support on only the spacelike separated points from it. With this in mind we introduce a Green's function which is non zero only for spacelike separated points:

$$(\Box - M^2)G(y,y') = \frac{1}{\sqrt{-g}}\delta^{d+1}(y-y')$$
$$G(y,y') = 0 \text{ for } y, y' \text{ not spacelike separated}. \tag{34}$$

We can now write $\phi$ using (34) and do two integrations-by-parts to get:

$$\phi(y) = \int d^{d+1}y'\, \phi(y')\, \delta^{d+1}(y-y')$$

$$= \int d^{d+1}y'\, \sqrt{-g}\, \phi(y')(\Box - M^2)\, G(y,y')$$

$$= \int d^d X n^\mu \left(\phi(X)\, \partial_\mu G(y,X) - G(y,X)\, \partial_\mu \phi(X)\right) + \int d^{d+1}y'\, \sqrt{-g}\, G(y,y')(\Box - M^2)\phi(y').$$

Here we have used our convention of using $y$ to denote bulk points and $X$ to denote boundary points. $n^\mu$ is the unit vector normal to the boundary.

The first integral can be evaluated from our knowledge of the boundary behavior of both the field and the Green's function.

We already know the field falls off as :

$$\lim_{r \to \infty} r^\Delta \phi(y) = \mathcal{O}(X). \tag{35}$$

The Green's function with one of its points at the boundary is a solution of the homogeneous Klein Gordon equation. Its boundary behavior is given by (13):

$$\lim_{r \to \infty} G(y,y') \to r^{d-\Delta}K(y,y') + r^{-\Delta}L(y,y'). \tag{36}$$

If we plug this back in the first integral only the leading order terms in $r$ will survive. Now if we put the interaction term to zero, the second integral vanishes by the equation of motion and we get back our CFT representation for the free field:[5]

$$\phi(y) = \int d^d X K(y;X)\mathcal{O}(X). \tag{37}$$

One can check that this gives the same smearing function as the one we had obtained earlier. Now we can add interactions. Then the second term will not be zero but subleading in $1/N$. We can solve it iteratively:

$$\phi(y) = \int d^d X K(y;X)\mathcal{O}(X) + \int d^{d+1}y'\, \sqrt{-g}\, G(y,y')(\Box - M^2)\phi(y')$$

$$= \int d^d X K(y;X)\mathcal{O}(X) \tag{38}$$

$$+ \frac{\lambda}{N}\int d^{d+1}y'\, d^d X'\, d^d X''\, \sqrt{-g}\, G(y,y')K(y';X')K(y';X'')\mathcal{O}(X')\mathcal{O}(X''). \tag{39}$$

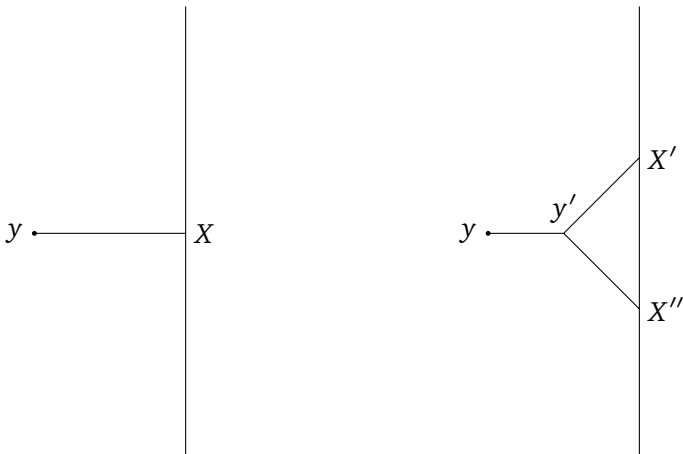

Figure 3: Diagrammatic representation of perturbative bulk reconstruction. The first diagram corresponds to the 0th order representation while the second corresponds to the first order correction in $1/N$.

We used (33) in the last step. This way an iterative series can be built order by order in $1/N$. The series can be represented diagrammatically:

The lines connecting bulk to boundary represent the smearing function $K(y;X)$ while bulk-to-bulk lines represent the spacelike Green's function $G(y,y')$.

## 3.2   $1/N$ corrections from Microcausality

A different strategy for implementing $1/N$ corrections comes from demanding microcausality in the bulk [27]. Let us sketch the idea.

We had obtained the CFT representation for the free field (30):

$$\phi^{(0)}(y) = \int dX K_\Delta(y;X)\mathcal{O}_\Delta(X), \tag{40}$$

where we have exhibited the operator dimensions explicitly. When we include interactions in the bulk($1/N$ corrections in the CFT) we would expect the representation to receive corrections of the form:

$$\phi(y) = \phi^{(0)}(y) + \phi^{(1)}(y) = \int dX K(y;X)\mathcal{O}(X) + \frac{1}{N}\sum_n \int dX' K_{\Delta_n}(y;X')\mathcal{O}_{\Delta_n}(X'), \tag{41}$$

where $\mathcal{O}_{\Delta_n}(X')$ are higher dimensional primaries. This is a natural guess because $\phi^1(y)$ falls off faster than $\phi^{(0)}(y)$ near the boundary and does not affect the extrapolate dictionary. Further, it doesn't affect the two-point function (primaries of different dimensions have vanishing two-point functions).

Now we can fix the $K_{\Delta_n}$ by demanding microcausality in the bulk. Microcausality is the property that two spacelike separated fields in the bulk commute: $[\phi(y),\phi(y')] = 0$ for spacelike separated $y,y'$. This a property we can demand of the CFT representation. For $1/N$ corrections we demand that this be satisfied in three point functions:

$$\langle 0|[\phi(y_1),\mathcal{O}(X_2)]\mathcal{O}(X_3)|0\rangle = 0. \tag{42}$$

---

[5]Actually we will not get back the same smearing function but the two can be shown to be equivalent using the ambiguities in smearing function discussed earlier.

Where we have taken two of the points to lie in the boundary for convenience. If one substitutes $\phi^{(0)}(y)$ in $\langle 0|[\phi(y_1), \mathcal{O}(X_2)]\mathcal{O}(X_3)|0\rangle$ it does not vanish. $\langle 0|\phi(y_1)\mathcal{O}(X_2)\mathcal{O}(X_3)|0\rangle$ and $\langle 0|\mathcal{O}(X_2)\phi(y_1)\mathcal{O}(X_3)|0\rangle$ both turn out to have singularities at different coordinate values. We can fix $K_n$ by demanding that including the first order correction to $\phi(y)$ cancels all the divergences and satisfies (42).

So we are demanding

$$\langle 0|\left[\int dX K_\Delta(y_1, X)\mathcal{O}(X) + \frac{1}{N}\sum_n \int dX' K_{\Delta_n}(y_1; X')\mathcal{O}_{\Delta_n}(X'), \mathcal{O}(X_2)\right]\mathcal{O}(X_3)|0\rangle = 0$$

$$\implies \int dX K_\Delta(y_1, X)\langle 0|[\mathcal{O}(X), \mathcal{O}(y_2)]\mathcal{O}(y_3)|0\rangle$$
$$+ \frac{1}{N}\sum_n \int dX' K_{\Delta_n}(y_1; X')\langle 0|[\mathcal{O}_{\Delta_n}(X'), \mathcal{O}(X_2)]\mathcal{O}(X_3)|0\rangle = 0. \quad (43)$$

Now all CFT three point functions are fixed (up to a constant) by conformal symmetry. This means that $\langle 0|[\mathcal{O}(X), \mathcal{O}(X_2)]\mathcal{O}(X_3)|0\rangle$ and $\langle 0|[\mathcal{O}_{\Delta_n}(X'), \mathcal{O}(X_2)]\mathcal{O}(X_3)|0\rangle$ are known. So we can use the above equation to solve for $K_{\Delta_n}$. This is another way of obtaining $1/N$ corrections.

This may seem to be giving a different result than the one we got from the Green's function approach (38). There the correction term was a single product of primaries $\mathcal{O}(X)\mathcal{O}(X')$ whereas here we have an infinite sum over products of local primaries. However, one can take the OPE $\mathcal{O}(X)\mathcal{O}(X')$ in (38) and obtain a tower of local primaries which matches precisely with what we have here.

A limitation of this method is that it doesn't directly generalize to higher-order corrections as higher point correlators are not fixed by symmetry alone.

# 4 Reconstruction of interacting gauge and gravitational fields

Let us consider a complex scalar field coupled to a $U(1)$ gauge field. The action is given by:

$$S = \int d^{d+1}x \sqrt{-g}\left(-D_\mu\phi^* D^\mu\phi - \frac{1}{4}F_{\mu\nu}F^{\mu\nu}\right), \quad (44)$$

where $D_\mu = \partial_\mu + iqA_\mu$, field strength $F_{\mu\nu} = \partial_\mu A_\nu - \partial_\nu A_\mu$ and $q$ is the coupling constant.

The gauge transformations in this theory are:

$$\phi(x) \to \phi(x)e^{-iqf(x)} \quad (45)$$
$$A_\mu(x) \to A_\mu(x) - \partial_\mu f(x). \quad (46)$$

Here the scalar field is local but not gauge-invariant. What are the gauge-invariant observables in this theory? A Wilson line attached to the scalar field is one example:

$$\Phi_W(x) = \phi(x)e^{iq\int_0^\infty ds A_\mu(x^\nu(s))\frac{dx^\nu(s)}{ds}}, \quad (47)$$

where the integration is over a curve that runs to some bulk point from the boundary: $s = \infty$ at the boundary while $s = 0$ at the target point $x$.

This is gauge-invariant but not local as one needs to know $A_\mu$ on an entire line that joins the boundary to the point $x$.

More generally we can construct gauge-invariant observables by introducing a function $g^\mu(x, x')$ [29]:

$$\Phi(x) = V(x)\phi(x)$$
$$= e^{iq \int d^d x' g^\mu(x,x') A_\mu(x')} \phi(x).$$

Under a gauge transformation parametrized given by $f(x)$:

$$V(x) \to V(x) e^{iq \int d^d x' g^\mu(x,x') \partial_\mu f(x')}. \tag{48}$$

Then $\Phi(x)$ is invariant under gauge transformations if

$$\partial_\mu g^\mu(x,x') = \delta^d(x-x').$$

Thus for any function $g^\mu(x,x')$ satisfying the above relation, we can construct a gauge-invariant observable.

We want to reconstruct $\Phi$ as a CFT operator. Again we take the coupling constant to be some $q = q'/N$. where $q'$ is some $\mathcal{O}(1)$ number. Then we can try to solve the equations of motion order by order in $1/N$. Note that we cannot directly implement microcausality as the gauge-invariant fields are non-local. However, we can obtain the modified microcausality relations and impose them. Alternately, we can demand that $\Phi$ transform suitably under bulk isometries. They would not transform like bulk scalars but one can figure out the transformation of, for instance, $\Phi_W$ from the transformation of $\phi$ and $A_\mu$. Imposing suitable transformation under isometries turns out to be enough to fix the corrections order by order. Typically, the corrections will include higher-order non-primary operators. This is expected because the gauge-invariant bulk field does not transform like a scalar. We refer the reader to [30] for details. An example of reconstructing gauge-invariant observables in a black hole background can be found in [31].

Similarly, when the gravitational field is considered we have diffeomorphism invariance, which is the gauge symmetry of Einstein's equations. This invariance tells us that coordinates themselves have no physical meaning. There can be no diffeomorphism invariant local observables, so when gravity is considered we would be forced to work with non-local observables.

Let us briefly discuss the construction of diffeo-invariant observables for gravitational fields. Given a boundary, it is straightforward to define such diffeomorphism invariant observables. An example of such an observable is one where a geodesic shoots out from some boundary point $X^i$. Then the field value $\phi(X^i, z)$– where $z$ is geodesic length calculated from the boundary – is a diffeomorphism invariant observable. This is so because diffeomorphisms at the boundary are not gauge symmetries of the theory.

Another class of diffeomorphism invariant observables are those are integrated over all of spacetime. An example would be vertex operators in worldsheet string theory.

We refer the reader to [32–36] for discussions on diffeomorphism invariant observables.

We emphasize that diffeomorphism invariance is a gauge symmetry of Einstein's equations (or Einstein's equations coupled to matter fields) and *not* of matter field equations in a fixed background. In other words, we have diffeomorphism invariance only when gravity is dynamical and not when we do field theory in a given background.[6]

In general relativity, matter fields also influence the background geometry through the right-hand side of Einstein's equation. This is called backreaction. When we work on a fixed background we neglect the backreaction of the matter on gravity. Diffeomorphism invariance only holds when backreaction is taken into account and one obtains the full solution of the coupled Einstein equations plus equation of motion for matter field.

---

[6]One can obtain a diffeomorphism-invariant formulation of a field theory in a given background by introducing auxiliary variables. This gives us parametrized field theories. The above comment is about when we do field theory without introducing such auxiliary variables.

For our program, the main consequence of this discussion is that when gravity is dynamical we have to work with diffeomorphism invariant observables. But when gravity is not dynamical, we should not use such observables. The construction of CFT representation is along similar lines as in gauge theories and was carried out in [37].

## 5 Reconstruction in AdS-Rindler and causal wedge reconstruction conjecture

In the last section, we saw that we can represent a scalar field at a bulk point $y$ as a non-local operator in the CFT using a smearing function $K(y;X)$ that has support at all boundary points $X$ spacelike separated from $y$. In this section, we will see that if we work with $K(y;X)$ that is a distribution rather than a function we could represent a bulk field in an even smaller region in the boundary.

### 5.1 Reconstruction in AdS-Rindler patch

The reader may be familiar with Rindler wedges in Minkowski space. The AdS-Rindler patch or wedge is an analogous region in AdS. In fact, it is the restriction of a Rindler wedge of the embedding d+2-dimensional Minkowski space to the AdS hyperboloid:

$$-X_0^2 + X_1^2 + \ldots X_d^2 - X_{d+1}^2 = -1 \,. \tag{49}$$

Consider a uniformly accelerated observer in the embedding space. Their worldline will be given by:

$$X_0 = \xi \sinh \tau \tag{50}$$
$$X_1 = \xi \cosh \tau \,. \tag{51}$$

Here $\tau$ is the time measured by the accelerated observer and $\frac{1}{\xi}$ is their acceleration. We take $\xi, \tau$ to be two of the coordinates in the AdS patch. We choose the rest of the coordinates $\chi, \Omega$ to satisfy (49)

$$X_{d+1} = \sqrt{\xi^2 + 1} \cosh \chi \tag{52}$$
$$X_2^2 + \ldots X_d^2 = (\xi^2 + 1) \sinh^2 \chi \,. \tag{53}$$

This gives us the metric on the AdS-Rindler patch:

$$ds^2 = -\xi^2 d\tau^2 + \frac{d\xi^2}{1 + \xi^2} + (1 + \xi^2)(d\chi^2 + \sinh^2 \chi \, d\Omega_{d-2}^2) \,. \tag{54}$$

The Rindler coordinates are related to the global coordinates. Here we give the relation for AdS$_3$ where the global coordinates are $(r, t, \theta)$ (as in (8)) and the Rindler coordinates are $(\xi, \tau, \chi)$. They are related as follows:

$$r^2 = \xi^2 \left[ \cosh^2 \chi + \sinh^2 \tau \right] + \sinh^2 \chi$$
$$\tan \theta = \frac{\sqrt{\xi^2 + 1} \sinh \chi}{\xi \cosh t}$$
$$\cos^2(t) = \frac{(\xi^2 + 1) \cosh^2 \chi}{\xi^2 \left[ \cosh^2 \chi + \sinh^2 \tau \right] + \cosh^2 \chi} \,. \tag{55}$$

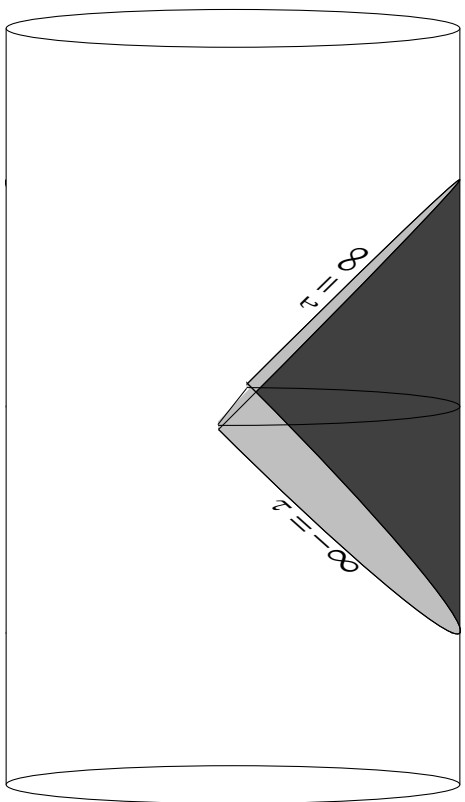

Figure 4: AdS/Rindler Wedge.

Note that $\tau \to \pm\infty$ as $t \to \pm\frac{\pi}{2}$. The Rindler patch covers only a part of the full AdS.

Now we can once again try to obtain a CFT representation for the bulk field by solving the wave equation. There are a few points of difference from the case of global AdS:

(i) Regularity at $r = 0$ is no longer imposed so the field modes are no longer quantized. One gets a continuum of modes. The upshot of this is that for a given Rindler patch one cannot alter the support of the integration region by adding modes, all modes contribute.

(ii) The smearing function diverges. The smearing function can be formally written as the Fourier transform of the AdS/Rindler mode functions $g_{k,\omega}$:

$$K(\chi, \tau, \xi; \tau', \xi') = \int dk \, d\omega \, e^{-i\omega\tau' + ik\xi'} g_{k,\omega}(\chi, \tau, \xi). \tag{56}$$

But the AdS/Rindler mode functions $g_{k,\omega}$ grow exponentially with $k$, which means that the smearing function diverges.

It was shown in [38] that this is not a problem as such. The smearing function in this case is not a function but it does make sense as a distribution. Integrating it with $\mathcal{O}(X)$ always gives sensible results.

The interesting point here is that for a field at any point of the Rindler wedge, the smearing distribution for the CFT representation has support only on the boundary of the Rindler wedge. It does not require information about the rest of the boundary.

This means that we can use the CFT Hamiltonian to reconstruct the bulk field in terms of the operators supported only on a part of the boundary Cauchy slice:

$$\phi(r, t, \Omega) = \int dt' d\Omega' K(r, t, \Omega, t', \Omega') e^{iHt'} \mathcal{O}(0, \Omega') e^{-iHt'}, \tag{57}$$

where the integration over $\Omega$ is only over the boundary of the Rindler wedge.

### 5.2   Causal wedge reconstruction conjecture

First let us define a causal wedge. Let R be a spatial subregion in the boundary. Then we can define something called the domain of dependence of R or $D[R]$. This is the set of all boundary points such that a null or a timelike curve passing through any of them must also intersect the region R. This is shown in the figure.

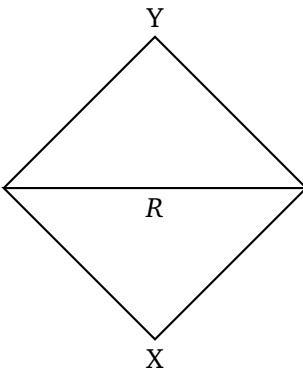

Figure 5: The domain of dependence of a boundary region $R$.

Now we define the causal wedge of R or $C[R]$ as the set of spacetime events in the bulk through which there exists a causal curve that starts and ends in $D[R]$. In the figure, this is the intersection of all bulk points lying to the causal future of X and all bulk points lying to the causal past of Y.

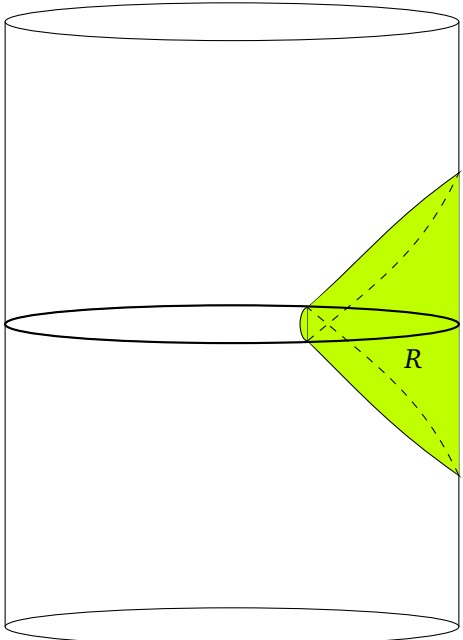

Figure 6: Causal Wedge of a boundary region $R$

The AdS-Rindler reconstruction can be extended to a more general class of bulk regions – the causal wedges of ball-shaped boundary regions (For AdS$_3$ 'ball-shaped regions' are simply intervals on the boundary). An AdS/Rindler chart can be defined on such wedges [38,39] and the above method applied. This result leads to the causal wedge reconstruction conjecture.

The causal wedge reconstruction conjecture holds that any field at any point in the causal wedge of any boundary region R in an asymptotically AdS spacetime can be reconstructed as

an operator in the boundary region $D[R]$. The intuitive explanation for this is that any point in $C[R]$ can be accessed by a causal observer starting from and returning to $D[R]$. But as the boundary theory is unitary, it already knows the information such an observer may bring. Thus the information about the entire causal wedge is already present in R. As noted, it is proved only for causal wedges of ball-shaped subregions in pure Anti-de Sitter spacetimes. For more general wedges in AdS, as well as for any causal wedges in more general asymptotically AdS spacetimes, this remains a conjecture.

It has been conjectured that an even bigger region than the causal wedge can be reconstructed from the information in R. This is the region known as the entanglement wedge of R. To define the region we first recall the covariant version of the Ryu-Takayangi proposal (usually called the HRT or Hubeny-Rangamani-Takayanagi proposal) for holographic entanglement entropy of a boundary region R [40, 41]. According to the HRT proposal, the entanglement entropy of R is equal to the area of the surface $\gamma_R$, which satisfies the following properties:
(i) It should be an extremal surface i.e a surface whose area is extremal under small variations.
(ii) It should be homologous to R.
(iii) The boundary of $\gamma_R$ should be the same as the boundary of R.

From the homology condition, we have that there exists a bulk region $H_R$ such that $\partial H_R = \gamma_R \cup R$. The domain of dependence of $H_R$ or $D[H_R]$ is called the entanglement wedge of R (denoted by $W[R]$).

$$W[R] = D[H_R]. \tag{58}$$

Generally, the entanglement wedge will contain the causal wedge. The entanglement wedge reconstruction conjecture holds that one can reconstruct fields in the entanglement wedge in the boundary of the wedge. We will not discuss entanglement wedge reconstruction in these lectures. We refer the reader to [42–45].

# 6 Scalar field reconstruction from symmetries

In this section, we discuss an alternate method of obtaining CFT representations for bulk fields given by Ooguri and Nakayama [46] based on earlier work by Miyaji et al [47] (see [48] for a related approach. A similar calculation also appeared in [49]). Unlike the previously described methods, this approach is based entirely on symmetry considerations. It is purely kinematical.

We will follow the original paper and use $(\rho, t, x^a)$ coordinates which are related to global coordinates as $\rho = \sinh r$ and $\Sigma_i x^a x_a = 1$. The metric in these coordinates given by

$$ds^2 = -\cosh^2 \rho \, dt^2 + d\rho^2 + \sinh^2 \rho \, d\vec{x}^2. \tag{59}$$

The key idea is to use the one-to-one correspondence between AdS isometries and the symmetries of the CFT. We start by asking how a CFT representation of a bulk scalar would transform under the conformal symmetries. In other words, we are asking what we should expect the following commutator to be:

$$[J^\mu, \phi_{CFT}(\rho, t, x^a)], \tag{60}$$

where $J^\mu$ are the generators of conformal symmetry in the CFT.

It is natural to expect the representation $\phi_{CFT}$ of a bulk field to transform under conformal symmetries in the same way bulk fields transform under bulk isometries. That is, there should be compatibility between boundary conformal transformations and bulk isometries.

Therefore they should satisfy the following commutation relations:

$$[J^\mu, \phi_{CFT}(\rho, t, x^a)] = \mathcal{J}^\mu \partial_\mu \phi_{CFT}(\rho, t, x^a), \tag{61}$$

where $\mathcal{J}^\mu$ is the killing field on AdS corresponding to the conformal symmetry generated by $J^\mu$. The Ooguri-Nakayama strategy is to try and find CFT operators $\phi_{CFT}$ that transform in this manner.

The conformal generators $J^\mu$ on the boundary of the AdS spacetime (which is a cylinder $\mathbb{R} \otimes S^{d-1}$) are the global Hamilltonian $H$ on $\mathbb{R}$, rotation generators $M_{ab}$ on the $d-1$-dimensional sphere, $P_a$ and $K_a$. The last two are respectively the translation and the special conformal generators when the cylinder $\mathbb{R} \otimes S^{d-1}$ is conformally mapped to $\mathbb{R}^d$.

We will need the following commutation relations:

$$[K_a, P_b] = 2\delta_{ab}H - 2iM_{ab}, [H, P_a] = P_a, [H, K_a] = -K_a. \tag{62}$$

Before we get to deriving $\phi_{CFT}$ from (61) let us give an explicit example.

Let us consider $\text{AdS}_2$. The conformal symmetry generators in this case are $K, P, H$. For each of them there is a corresponding isometry in the bulk. They are:

$$\mathcal{H} = \partial_t.$$
$$\mathcal{K} = \frac{1}{2}\tanh\rho e^{-it}\partial_t + ie^{-it}\partial_\rho.$$
$$\mathcal{P} = \frac{1}{2}\tanh\rho e^{it}\partial_t - ie^{it}\partial_\rho.$$

The condition (61) then translates to the following conditions in $\text{AdS}_2$:

$$[H, \phi_{CFT}(\rho, t)] = \partial_t \phi_{CFT}(\rho, t).$$
$$[K, \phi_{CFT}(\rho, t] = (\frac{1}{2}\tanh\rho e^{-it}\partial_t + ie^{-it}\partial_\rho)\phi_{CFT}(\rho, t).$$
$$[P, \phi_{CFT}(\rho, t)] = (\frac{1}{2}\tanh\rho e^{i}t\partial_t - ie^{it}\partial_\rho)\phi_{CFT}(\rho, t).$$

Now we return to $\text{AdS}_{d+1}$ and start trying to construct operators that satisfy (61). First, we reconstruct the scalar field at the origin. As one can check, the following transformations leave the scalar field at the origin invariant:

$$[M_{ab}, \phi_{CFT}(0)] = 0. \tag{63}$$
$$[P_a + K_a, \phi_{CFT}(0)] = 0. \tag{64}$$

The second condition can be immediately checked to be true in the $\text{AdS}_2$ example above. Let us define the state $\phi_{CFT}(0)|0\rangle = |\phi_{CFT}(0)\rangle$. Then (63) and (64) translate to

$$M_{ab}|\phi_{CFT}(0)\rangle = 0; (P_a + K_a)|\phi_{CFT}(0)\rangle = 0. \tag{65}$$

To solve this problem we start from a primary scalar of dimension $\Delta_\phi$:

$$M_{ab}|\mathcal{O}\rangle = 0; K_a|\mathcal{O}\rangle = 0; H|\mathcal{O}\rangle = \Delta_\phi. \tag{66}$$

We write down an ansatz for a solution to (65) by adding all the descendants of this primary to it with arbitrary coefficients.

$$|\phi_\Delta\rangle\rangle = \sum_{n=0}^{\infty}(-1)^n a_n (P^2)^n |\mathcal{O}\rangle. \tag{67}$$

Now we can impose (65) on the above equation and solve for $a_n$. This gives the following result:

$$a_n = \prod_{k=1}^{n} \frac{1}{4k\Delta_\phi + 4k^2 - 2kd}. \tag{68}$$

This defines a scalar field at the origin $|\phi_\Delta\rangle$. We can reconstruct other scalar fields starting from primaries of different dimensions. A general scalar field at the origin will be given by:

$$|\psi_{CFT}(0)\rangle = \sum_\Delta b_\Delta |\phi_\Delta\rangle. \tag{69}$$

We can shift the field to any other point on the bulk using the generators that don't leave the origin invariant:

$$|\phi_{CFT}(\rho, t, x^a)\rangle = e^{-iHt} e^{\rho(P_a - K_a)x_a} |\phi_\Delta\rangle. \tag{70}$$

Note that so far we have made no reference to dynamics. We have reconstructed a general scalar field.

But as we will now show, the state obtained in (70) is a solution to the Klein Gordon equation.

To see this, we introduce the quadratic Casimir operator of the CFT. This operator commutes with all the conformal generators:

$$C_2 = \frac{1}{2} M_{\mu\nu} M^{\mu\nu} - K_\nu P^\mu - dD + D^2, \tag{71}$$

where $d$ is the number of dimensions. It is easy to check using (66) that the primary state $|\mathcal{O}\rangle$ is an eigenstate of this operator with eigenvalue $\Delta_\phi(\Delta_\phi - d)$. But since $C_2$ commutes with all the generators that appear in (70), acting it on $|\phi_{CFT}(\rho, t, x^a)\rangle$ we have

$$C_2 |\phi(\rho, t, x^a)\rangle = \Delta_\phi(\Delta_\phi - d)|\phi(\rho, t, x^a)\rangle. \tag{72}$$

Defining an operator $\phi_{CFT}(\rho, t, x^a)$ by $\phi_{CFT}(\rho, t, x^a)|0\rangle = |\phi_{CFT}(\rho, t, x^a)\rangle$ we can write the above equation as

$$[C_2, \phi_{CFT}(\rho, t, x^a)] = \Delta_\phi(\Delta_\phi - d)\phi_{CFT}(\rho, t, x^a). \tag{73}$$

But from (61) and using the explicit form of $C_2$ given in (71) we can translate the commutators of generators to actions of bulk isometries.

This turns out to give us:

$$\Box \phi_{CFT} = M^2 \phi_{CFT}, \tag{74}$$

where $\Box$ is the usual box operator (also known as the Laplace-Beltrami operator) on AdS and $M^2 = \Delta_\phi(\Delta_\phi - d)$. This is consistent as the Laplace-Beltrami operator commutes with all the AdS isometries.

So each CFT state $|\phi_{CFT}\rangle$ is dual to a solution to a different free field equation in the bulk.

Finally, let us sketch how one can go from the HKLL representation to the one we just obtained. We will be very schematic and refer the reader to Appendix A.5 of [50] for details.

Let us work in AdS$_2$. We start by acting on the vacuum with $\phi_{CFT}(0)$, the HKLL representation of the field at the origin. This gives us :

$$\int dt\, K(0; t)\mathcal{O}(t)|0\rangle = \int dt\, K(0; t) \sum_n \frac{t^n}{n!} (\partial_t)^n \mathcal{O}(t)|_{t=0}|0\rangle$$

$$= \sum_n \left( \frac{1}{n!} \int dt\, t^n K(0; t) \right) P^n |\mathcal{O}\rangle.$$

Here we did a Taylor expansion in the first step. For AdS$_2$ it turns out that the integral $\int dt K(0;t)t^n$ vanishes for odd n. Then we get exactly the form

$$|\phi_{CFT}\rangle\rangle = \sum_{n=0}^{\infty}(-1)^n a_n (P^2)^n|\mathcal{O}\rangle, \tag{75}$$

where

$$(-1)^n a_n = \int dt K(0;t)\frac{t^n}{n!}. \tag{76}$$

One can check that they match exactly. The ambiguity in the smearing function discussed earlier corresponds to the invariance of (67) under the transformation $a_n \to a_n + b_n$ where

$$\sum_{n=0}^{\infty}(-1)^n b_n (P^2)^n|\mathcal{O}\rangle = 0. \tag{77}$$

Thus the two representations are equivalent.

Further progress along the line of the Ooguri-Nakayama approach was made in [50, 51]. In [50] local states in a BTZ background were constructed. An extension of this approach to find CFT representation for fields which transform as scalars under asymptotic symmetries in AdS$_3$ was given in [52, 53].

A limitation of this approach is that it is purely kinematical. There is no way to incorporate dynamical information in the CFT representation constructed in this way.

# 7 Challenges to bulk reconstruction

In this section, we review the challenges to the bulk reconstruction program. First, there is the issue of going to the finite $N$ regime. So far we have worked in the infinite $N$ regime which is dual to semiclassical bulk physics. But ultimately we would need to understand the finite $N$ regime which is dual to bulk quantum gravity. Even in the large $N$ regime, the bulk reconstruction program faces challenges in the presence of horizons. In the following sections, we discuss these issues.

## 7.1 Challenges at finite $N$

At finite $N$, the semiclassical picture of the bulk with local field theories living on some given background is expected to break down.

One can see this from black hole thermodynamics. From renormalization group wisdom, we expect any field theory to flow to a conformal field theory at very high energies.[7] If quantum gravity is a local field theory at all energies, we would expect it too to flow to a CFT. Then its entropy must scale like a (d+1)-dimensional CFT as $E^{\frac{d}{d+1}}$.

Now if high enough energy is concentrated in a bulk region it results in the formation of a black hole. The high-energy spectrum of gravity is therefore dominated by black holes. Consequently, we would expect the entropy in the quantum gravity theory to be dominated by entropy coming from black hole microstates.

But black hole entropy is given by the Bekenstein-Hawking formula

$$S_{BH} \approx \frac{A}{G} = \frac{r_s^{d-1}}{G} = \frac{M^{\frac{d-1}{d}}}{G^{1/d}}, \tag{78}$$

---

[7]For those not familiar with the renormalization group, this simply means that at very high energies any field theory behaves like a CFT.

where $r_s$ is the Schwarzschild radius given by $r_s^d \approx GM$ for AdS-Schwarzschild black holes in the limit of large Schwarzschild radius.

This is much smaller than required for a d+1-dimensional CFT.

However it is perfect for a $d$ dimensional CFT! The entropy of a d-dimensional CFT with energy $E$ and central charge $N^2$ is given by:

$$S = N^{\frac{2}{d}} E^{\frac{d-1}{d}} \, . \tag{79}$$

Using $N^2 = 1/G$ and identifying black hole mass with the energy, this matches exactly.

So we see that the local field theory picture in the bulk over-counts degrees of freedom. If we try to re-create local field theory from CFT, the process must break down. In the large $N$ limit, there is no problem. A hand-waving way of seeing this is that in this limit the boundary entropy also blows up. One can make the agreement between the entropy of a bulk local theory and a large $N$ CFT precise [16].

So this method of reconstructing bulk observables must fail at finite $N$. At present, we don't know how to go to the finite $N$ regime.

## 7.2 Challenges to bulk reconstruction behind the horizon

Even in the large $N$ limit one faces some severe challenges when one tries to reconstruct bulk fields behind the horizon of a black hole.

As we will see, the HKLL construction fails beyond the horizon of a collapsing black hole. Papadodimas and Raju have a proposal for the construction of operators beyond the horizon in terms of state-dependent 'mirror operators'. These operators satisfy all the properties expected of a bulk field mode beyond the horizon. But their definition depends on the particular state of the black hole.

On the other hand, there is the Marolf-Wall paradox which says that if bulk fields everywhere (including behind the horizon) can be represented as linear CFT operators, one can show for the case of an asymptotic AdS space with more than one boundary that the boundary CFTs cannot capture full information about the bulk. There can be more than one bulk geometry dual to the same boundary state.

In this section, we will discuss the state-dependence proposal of Papadodimas and Raju and the Marolf-Wall paradox.

### 7.2.1 Bulk reconstruction in collapsing black holes.

Eternal black holes pose no problem of principle for HKLL construction. Outside the horizon, the bulk fields are reconstructed as before as operators in either the left or the right boundary CFT. The field behind the horizon will be represented as a sum of operators of the CFTs of both the boundaries. This is made clear in the figure below.

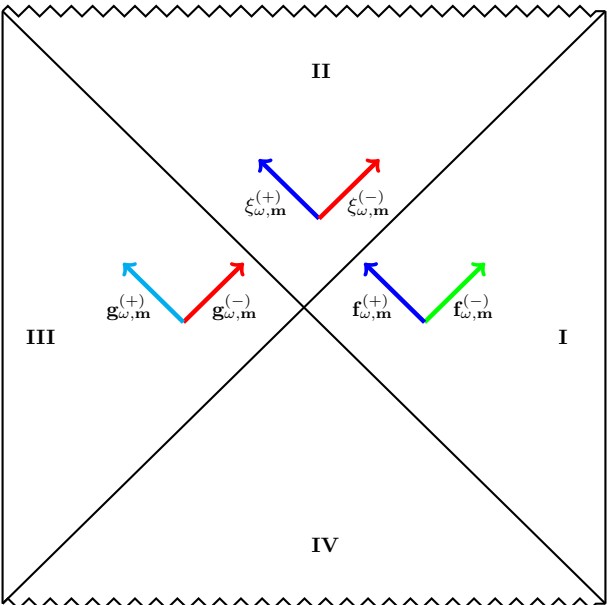

Figure 7: Bulk reconstruction in an eternal black hole. $f^{(\pm)}_{\omega,m}$ are the solutions to the wave equation in region I which behave like plane waves near the horizon. A linear combination of them is the normalizable mode. The CFT representation of a field in region I will be an operator on the right boundary. In region II, we have modes coming in both from regions I and III. To obtain the CFT representation one continues back these modes to the boundary. This gives us a sum of two operators – one on the left CFT and one on the right.

But it fails for collapsing black holes. First, let us understand why the HKLL construction fails beyond the horizon for collapsing black holes. We consider a black hole formed from the collapse of a null shell.

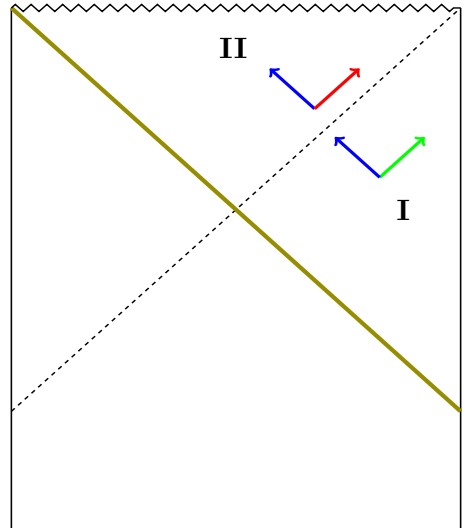

Figure 8: Bulk reconstruction fails for a collapsing black hole. The left moving mode (in blue) can be continued back to the boundary but the right moving mode (in red) inside the horizon when continued back collides with the infalling null shell (in olive) at transplanckian energies

In principle, we can reconstruct the field at any point outside the horizon by solving the

field equations as before. Inside the horizon, the left moving modes pose no problem of principle either. They can be continued back to the boundary. It is the right moving modes inside the horizon that are problematic. To carry out bulk reconstruction beyond the horizon we would need to continue these modes back to the origin, reflect them through the origin and continue back till the boundary. However, these modes will get blue-shifted when continued backward. For sufficiently late modes this means that when continued back they will collide with the collapsing matter at very high (greater than Planck scale) center-of-mass energies. Classical field equations break down at that point. This 'transplanckian' issue is why we cannot construct the bulk field beyond the horizon.

While HKLL construction fails, Papadodimas and Raju [54–56] have argued that one can still find boundary representations of bulk fields, but they will be 'state-dependent' operators.

We will now sketch the mirror operator construction of Papdodimas and Raju in a simplified manner. Our presentation will gloss over subtleties for clarity and we refer the reader to the original papers for an accurate presentation. We also refer to the original papers for a discussion about how the different versions of the firewall paradox are evaded by these mirror operators (indeed this was the motivation behind their construction).

Now let us consider a black hole formed from collapse. We consider the black hole to be big enough not to evaporate.[8] We assume that sufficient time has passed after the collapse so that fluctuations have died down and the black hole is approximately in equilibrium.

At this late time the metric will resemble the eternal black hole geometry:

$$ds^2 = -f(r)dt^2 + \frac{1}{f(r)}dr^2 + r^2 d\Omega^2 \,, \tag{80}$$

where

$$f(r) = r^2 + 1 - \frac{c_d GM}{r^{d-2}} \,, \tag{81}$$

$c_d$ is a dimension dependent constant. The horizon is at $r_0$ where $f(r_0) = 0$.

We consider a massless scalar field in this background. We would like to get a CFT representation for all the modes of this field. We introduce the tortoise coordinates $r_*$:

$$\frac{dr_*}{dr} = \frac{1}{f(r)} \;\; ; r_* = 0 \text{ at } r = 0 \,. \tag{82}$$

We can solve the wave equation in this background and find the modes. Sufficiently close to the horizon they act like plane waves (the near-horizon region of any black hole is approximated by Rindler space). In what follows we are suppressing the angular dependence, which does not play a crucial role.

Outside the horizon we have:

$$\phi(t, r_*, \Omega) = \Sigma_{l, \vec{m}} \int \frac{d\omega}{\sqrt{\omega}} \left( a_{\omega, l, \vec{m}} e^{-i\omega(t+r_*)} + e^{i\delta_\omega} b_{\omega, l, \vec{m}} e^{-i\omega(t-r_*)} \right) Y_{l, \vec{m}}(\Omega) + h.c \,. \tag{83}$$

The origin of the phase $e^{i\delta_\omega}$ is the normalizability condition – modes outside the horizon must vanish at the boundary and it turns out that only a particular linear combination of the two modes above vanishes at the boundary.

Inside the horizon:

$$\phi(t, r_*, \Omega) = \Sigma_{l, \vec{m}} \int \frac{d\omega}{\sqrt{\omega}} \left( a_{\omega, l, \vec{m}} e^{-i\omega(t+r_*)} + \tilde{b}_{\omega, l, \vec{m}} e^{i\omega(t-r_*)} \right) Y_{l, \vec{m}}(\Omega) + h.c \,. \tag{84}$$

---

[8]Remember that AdS is like a box, so only black holes so small that their lifetimes are smaller than the time taken for radiation from the black hole to be reflected back from the boundary can evaporate.

All modes other than $\tilde{b}$ can be represented as CFT operators using HKLL construction. To find a representation for $\tilde{b}$ we will first learn how it behaves inside correlators. Then we can try to find an operator in the CFT which reproduces the same correlators.

In what follows we will only consider the $l = 0$ mode and denote $b_{\omega,l,\vec{m}}$ with $l = 0$ as simply $b_\omega$.

Now for short distances the two point function behaves as:

$$\langle \phi(x)\phi(y) \rangle \approx \frac{1}{\left[ g^{\mu\nu}(x-y)_\mu(x-y)_\nu \right]^{\frac{d-1}{2}}} . \tag{85}$$

We can choose the points $x, y$ to be both inside, both outside or on inside and one outside of the horizon. Substituting (84) and (83) in (85) for each of these cases we obtain the following correlation functions:

$$\langle \tilde{b}_{\omega'} \tilde{b}_\omega^\dagger \rangle = \frac{1}{1 - e^{-\beta\omega}} \delta(\omega - \omega') \tag{86}$$

$$\langle \tilde{b}_\omega b_{\omega'} \rangle = \frac{e^{-\frac{\beta\omega}{2}}}{1 - e^{-\beta\omega}} \delta(\omega - \omega') . \tag{87}$$

The CFT operator that represents some $\tilde{b}$ must replicate this behavior. The aim then is to find such CFT operators.

The key idea is to note that we need a CFT operator that reproduces only those bulk correlation functions of a given $\tilde{b}_\omega$ operator which correspond to all reasonable experiments that a bulk observer might do.

A reasonable experiment is one that can be described by effective field theory in the bulk. An example of an unreasonable experiment would be one where we localize so much energy in a small region that a black hole gets formed.

The first step is to obtain this set of operators which describe such reasonable experiments. To do this one first discretizes the modes by introducing a time band $[-T_b, T_b]$:

$$b_\omega = \frac{2\pi}{\sqrt{T_b}} \int_{-T_b}^{T_b} dt\, \mathcal{O}(t) e^{i\omega t} . \tag{88}$$

Then one consider the set of polynomial operators spanned by the set of monomials $\{b_{\omega_1}, b_{\omega_1}b_{\omega_2}, ...., b_{\omega_1}....b_{\omega_n}\}$. Polynomials obtained by taking linear combinations of these can be considered to describe reasonable experiments provided that the following conditions are met by each monomial:

(i) They should not have so much energy that they form a black hole:

$$\sum_i \omega_i \ll \mathcal{O}(N^2) . \tag{89}$$

(ii) There should not be too many insertions, which can also lead to the breakdown of effective field theory. This gives a condition on the number of insertions $k$:

$$k \ll \mathcal{O}(N^2) . \tag{90}$$

The set of polynomials spanned by the monomials satisfying the above conditions is denoted as $B_{eff}$. To this one also adds $B_H$ the set of polynomials in small powers of the CFT Hamiltonian $B_H = \text{span}\{H, H^2, ...H^n\}; n \ll N$. This gives us the set of operators $B$.

The upshot of these two conditions is that $D_B \ll \mathcal{O}(e^{N^2})$, where $D_B$ is the dimension of $B$. With these restrictions, $B$ forms the set of all reasonable experiments. It can be thought of as an

approximate algebra (sometimes called a small algebra) of effective field theory observables. It is only an approximate algebra because some compositions of elements of the set $B$ won't satisfy the restrictions above and take one outside the set $B$.

Once we have the set of all reasonable experiments, the next step is to choose an appropriate state for the black hole. Here we introduce the notion of 'equilibrium states' – states which are dual to black holes that are in approximate equilibrium. The defining property of equilibrium states is that the elements of the small algebra will have approximately (i.e up to $1/N$ corrections) thermal correlators for these states. Note that there can be more than one equilibrium state dual to a given black hole geometry.

Now we choose such an equilibrium state $|\psi\rangle$. We form the linear $H_{|\psi\rangle}$ by acting on $|\psi\rangle$ by operators in $B$.

$$H_{\psi} = B|\psi\rangle = \text{span}\left\{\sum_p b_p B_p|\psi\rangle\right\}, \tag{91}$$

where $B_p$ are elements of $B$. This can be thought of as the subspace corresponding to the effective field theory near the equilibrium state $|\psi\rangle$. It is sometimes called the little Hilbert Space. It can be shown that the dimension of $H_{\psi}$ is also $D_B \ll e^{N^2}$.

Now we are ready to define the CFT operators that represent $\tilde{b}$ modes. These are defined by their action on $\psi$:

$$\tilde{b}_{\omega}B_{\alpha}|\psi\rangle = e^{-\frac{\beta\omega}{2}}B_{\alpha}b_{\omega}^{\dagger}|\psi\rangle$$
$$\tilde{b}_{\omega}^{\dagger}B_{\alpha}|\psi\rangle = e^{\frac{\beta\omega}{2}}B_{\alpha}b_{\omega}|\psi\rangle. \tag{92}$$

For any $B_{\alpha} \in B$. The $b$ that appears here is the CFT representation of the $b$ modes in the bulk. The $\tilde{b}$ are called mirror operators as they mirror the action of $b$ modes.

How do we know that one can always find $\tilde{b}_{\omega}$ and $\tilde{b}_{\omega}^{\dagger}$ that satisfy (92)? As all $B_{\alpha}|\psi\rangle$ are linearly independent, (92) defines the operators $\tilde{b}_{\omega}$ and $\tilde{b}_{\omega}^{\dagger}$ by a set $D_B$ equations each. But the approximate dimension of the full Hilbert space is $e^{N^2}$ so the operators $\tilde{b}_{\omega}$ and $\tilde{b}_{\omega}^{\dagger}$ can be thought of as a $e^{N^2} \times e^{N^2}$ matrices. But as $D_B \ll e^{N^2}$, solutions can always be found for the above set of equations.

Such solutions may not be unique but that's not an issue as the definition completely specifies the action of mirror operators within the little Hilbert Space. What happens beyond the little Hilbert Space is irrelevant for our purposes.

Now let us check if the mirror operators in the CFT do indeed reproduce the correct correlators.

We have:
$$\tilde{b}_{\omega}^{\dagger}\tilde{b}_{\omega'}|\psi\rangle = e^{-\frac{\beta\omega}{2}}\tilde{b}_{\omega}^{\dagger}b_{\omega'}^{\dagger}|\psi\rangle = b_{\omega'}^{\dagger}b_{\omega}^{\dagger}|\psi\rangle. \tag{93}$$

Where we have used the definitions (92) at each step. Then

$$\langle\tilde{b}_{\omega}^{\dagger}\tilde{b}_{\omega'}\rangle = \langle b_{\omega'}^{\dagger}b_{\omega}\rangle. \tag{94}$$

So these are indeed the correct correlators. One can also check from the definition that for any state in the little Hilbert space $B_{\alpha}|\psi\rangle$, it is true that

$$[\tilde{b}_{\omega}, b_{\omega}]B_{\alpha}|\psi\rangle = 0 \tag{95}$$

and

$$[\tilde{b}_{\omega}, \tilde{b}_{\omega}^{\dagger}]B_{\alpha}|\psi\rangle = B_{\alpha}|\psi\rangle. \tag{96}$$

The correct commutation relations are recovered, but only within the little Hilbert space.

In general, $\tilde{b}$ operators won't commute with $b$ operators. This means that locality is lost. However, they will commute within the little Hilbert Space so we still have a local effective field theory.

Let us discuss the state dependence of this construction. The mirror operators that we obtained were for one equilibrium state $|\psi_1\rangle$. We could have denoted the operator as $b_{|\psi_1\rangle}$. If we started with a different equilibrium state $|\psi_2\rangle$ which corresponds to the same geometry we would have obtained a different little Hilbert Space and a different mirror operator $b_{|\psi_2\rangle}$ for the same bulk modes. So to know which operator to use to describe the modes behind the horizon, one has to know which exact equilibrium state one is in. It is not enough to know the geometry.

This is in contrast to the HKLL construction. HKLL construction also has a kind of 'state-dependence' in that they depend on the background geometry, and different background metrics correspond to different CFT states. So when we obtain the HKLL representation in pure AdS, it only holds for the vacuum state in the CFT (and excitations around it). A different CFT state would have a different bulk geometry dual to it and we would have a different HKLL representation for that state. However, knowing the geometry is enough. But for mirror operators, that is not the case.

It has been argued that such state-dependent operators form a non-linear modification of quantum measurement theory. We refer the reader to [57] for a discussion on this.

## 7.3 Marolf-Wall paradox: AdS $\neq$ CFT?

In this last section, we review the paradox posed by Marolf and Wall [58]. The Marolf-Wall argument concerns asymptotically AdS geometries with multiple boundaries.

It is generally believed that if an asymptotic AdS geometry has $n$ boundaries then the dual to this geometry is an entangled state of n non-interacting CFTs. This is a very reasonable belief as the two boundaries cannot possibly interact unless there is a traversable wormhole. The most well-understood example of this is the two-sided eternal black hole, which is dual to a particular entangled state in the CFT known as the thermofield double state.

The Marolf-Wall argument shows that if semiclassical bulk observables can be translated to linear operators in the CFT, then an asymptotically AdS spacetime with more than one boundary cannot be dual to a CFT state. The essence of the argument is that there can be more than one bulk dual to an entangled CFT state. In other words, the map between CFT states and bulk duals cannot be one-to-one.

Let us review their argument for the thermofield double state, which is a state in two entangled non interacting CFTs, which we will call left and right CFT for convenience. We introduced this state in (4):

$$|\psi_{\text{TFD}}\rangle = \frac{1}{Z(\beta)} \sum_E e^{\frac{-\beta E}{2}} |E\rangle_L |E\rangle_R. \tag{97}$$

This is dual to the two-sided eternal black hole. But Marolf and Wall argued that there is another dual. To see this, note that by the AdS/CFT correspondence each CFT in $\text{CFT}_L \otimes \text{CFT}_R$ is dual to a bulk theory living in a one-sided asymptotically AdS geometry. For instance any factorized state in $\text{CFT}_L \otimes \text{CFT}_R$ can always be interpreted as a tensor product of two disconnected bulk geometries, which can be possibly quantum (which is to say, they can have large fluctuations). For instance, $|0\rangle|0\rangle$ is dual to a disconnected pair of pure AdS geometries. Similarly, any state $|E\rangle|E\rangle$ will be dual to some disconnected pair of (quantum) geometries, each of which is dual to an energy eigenstate of a CFT. Now we should also be able to interpret the thermofield double state as a superposition of such disconnected pairs of asymptotically AdS geometries.

So we have two possible bulk duals for the thermofield double state.

But can they really be distinguished by an experiment? Could it be that they are different mathematical representations of the same physical state? In which case no experiment could ever distinguish between them. Marolf and Wall argued that the answer is no. They proposed an experiment whose result depends on the choice of bulk dual.

In the experiment, we consider an observer Alice who starts from one of the boundaries (which we will call the left boundary for convenience)and moves in. We can define such an Alice for both the eternal black hole and the disconnected geometries.

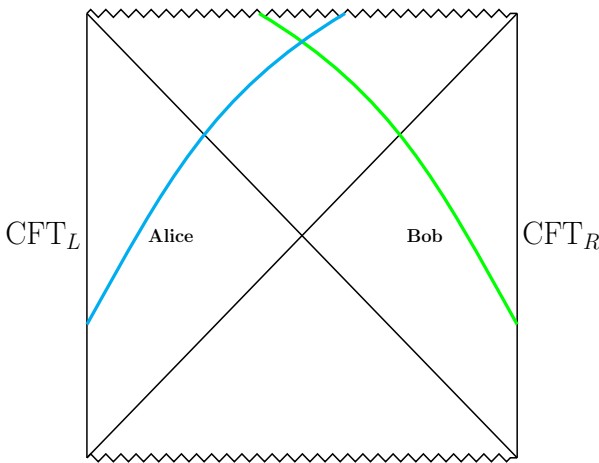

Figure 9: Alice and Bob travel from the left and right CFTs towards the horizon.

We will describe Alice by unitary an operator $e^{iA}$. This point may be confusing so let us digress and elaborate on this point.[9] The reader can skip this discussion in the first reading.

To describe an observer in an asymptotic region what we need is a localized wave packet at the asymptotic infinity. Now in a small enough region in the boundary, all the correlators will look close to the correlators in pure AdS, irrespective of the interior geometry. Therefore a localized 'Alice' wave packet can be constructed by some operator acting on the CFT vacuum which would look like:

$$O_A|0\rangle = \sum_n \int dk_1...dk_n f(k_1,...,k_n) a^\dagger_{k_1}..a^\dagger_{k_n}|0\rangle. \tag{98}$$

Where $a^\dagger_\omega$ are CFT representations of creation operators for the pure AdS. From the previous section, we already know how to obtain them. But this is not a unitary operator. However, we can always find a unitary $U$ which mimics this operator acts exactly the same way in the vacuum:

$$\langle i|U_A|0\rangle = \langle i|O_A|0\rangle \tag{99}$$

for some basis $|i\rangle$. This is all we need to represent a localized asymptotic observer. But we still have complete freedom to fix all other matrix elements $\langle i|U|j\rangle$. One can show that it is always possible to find a unitary operator that satisfies this property.

Let us return to the experiment. First, we consider the case where the dual is an eternal black hole. We consider the black hole to be large enough so that a semiclassical description holds inside the horizon (except near the singularity). Now in an eternal black hole, we can define another observer Bob (defined by another unitary operator $e^{iB}$) who starts from the right boundary. It can be arranged so that Alice and Bob meet behind the horizon.

---

[9]We thank Aron Wall for explaining this point to us in detail.

Now we ask the question 'Does Alice meet Bob when she jumps inside a black hole?' This is a well-defined question in the bulk. In an eternal black hole, the probability of Alice meeting Bob is close to one. We can write this schematically as:

$$_{\mathbf{bh}}\langle\text{Does Alice meet Bob when both are created appropriately?}\rangle_{\mathbf{bh}} \approx 1 \qquad (100)$$

Let us translate this to the CFT. In the CFT the answer to this question is given by the projector $P$ which projects on to the states where Alice meets Bob. Then we have:

$$\langle\psi_{\text{TFD}}|e^{-i(A+B)}Pe^{i(A+B)}|\psi_{\text{TFD}}\rangle \approx 1\,. \qquad (101)$$

If we don't create the operator Bob on the right boundary we should have

$$\langle\psi_{\text{TFD}}|e^{-iA}Pe^{iA}|\psi_{\text{TFD}}\rangle \approx 0\,. \qquad (102)$$

Now let us ask the same question for the other bulk dual to the thermofield double state, the superposition of disconnected pairs of one-sided geometries. This is again an operationally well-defined question in $\mathbf{bulk}_L$ where Alice lives. By AdS/CFT correspondence we should be able to answer this question by a projector $P_L$ which lives in CFT$_L$. This is an important point – by AdS/CFT correspondence the answer to the question 'does Alice meet Bob' in a one-sided bulk must be given by a projector in a single CFT.

In this case we can calculate the result directly in the boundary theory:

$$\langle\psi_{\text{TFD}}|e^{-i(A+B)}P_L e^{i(A+B)}|\psi_{\text{TFD}}\rangle = \langle\psi_{\text{TFD}}e^{-iA}P_L e^{iA}|\psi_{\text{TFD}}\rangle \approx 0\,. \qquad (103)$$

The second step follows because $P_L$ is an operator in the left CFT and commutes with operators from the right CFT. The probability is not exactly zero because quantum fluctuations can always create a Bob. To make this probability really small we can arrange the experiment to be so that Bob carries some qubit which Alice will measure. The probability that a Bob-like wave-packet along with a particular qubit gets created by quantum fluctuations is vanishingly small(see the discussion in Appendix A of [58]).

So for the bulk dual (which we label as '$\mathbf{dc}$'), this translates to;

$$_{\mathbf{dc}}\langle\text{Does Alice meet a Bob when both are created appropriately?}\rangle_{\mathbf{dc}} \approx 0\,. \qquad (104)$$

So we seem to have arrived at a contradiction. A well-defined question in the bulk elicits different answers from the same CFT state depending on what bulk interpretation we use. The two possible bulk duals to the thermofield double state can be distinguished by an operationally well-defined experiment.

This means that one cannot distinguish between these two bulk geometries from the CFT – the same CFT state is dual to both. Therefore the general bulk theory which contains both these states can't be dual to a CFT ⊗ CFT. Instead, it should be dual to CFT ⊗ CFT ⊗ S, where S is the space that contains this additional information which can distinguish between the two states.

There are three possible ways out of the Marolf-Wall paradox which have been suggested in the literature.

The first comes from state dependence, which says that one can't construct fields behind the horizon as linear operators in the CFT. If one can't construct $P_L$ as a linear operator in the bulk then (103) does not hold. Even if there is a very small probability of Alice meeting Bob in each factorized state $|E\rangle_L|E\rangle_R$, the sum over states may yield a number close to one if $P_L$ is nonlinear. If observables behind the horizon can't be represented by state independent operators as has been argued by Papadodimas and Raju, that would be a way out of the Marolf-Wall paradox.

Another way out has been suggested by Jafferis [59], who has argued that the kind of observables involved in describing this experiment may not be good bulk observables. Jafferis has argued that good bulk observables must be non-perturbatively diffeomorphism invariant and these observables do not satisfy that criterion.

A different point of view on the Marolf-Wall paradox [60] is that one should not interpret the argument to imply that there are superselection sectors. Rather, the more natural interpretation of the argument is that a state in CFT ⊗ CFT can have two bulk duals which differ in their operator dictionaries. Note that the same bulk question was answered by different projection operators in the CFT in the two cases. This means that the bulk-boundary operator dictionaries are different in the two cases. There is no contradiction in the CFT and one could argue that a single CFT state having multiple bulk duals with differing operator dictionaries is not a paradox in itself.

# 8 Conclusion

In these lectures, we reviewed the program of completing the bulk-boundary dictionary. We reviewed the HKLL construction in Anti-de Sitter spacetime and obtained the smearing function for free and interacting theories. We saw that for an AdS-Rindler patch the smearing function is a distribution instead. However, using a distribution one can obtain a representation smeared over a smaller boundary region. We discussed bulk reconstruction from symmetries.

We also reviewed challenges to bulk reconstruction. We only understand bulk reconstruction at large $N$, the case of finite $N$ (i.e quantum gravity in the bulk) remains a challenge. Even at large $N$, the existence of a horizon poses challenges. We saw that for black holes formed from collapse the HKLL procedure fails. However a prescription for bulk reconstruction in terms of mirror operators exists, which we reviewed. Finally, we reviewed the Marolf-Wall paradox which challenges the idea that the AdS/CFT dictionary is one-to-one.

# Acknowledgements

It's a pleasure to thank Alok Laddha for introducing me to this topic and for many valuable discussions over the years. I am extremely grateful to Daniel Harlow, Daniel Kabat, Gilad Lifschytz and Aron Wall for patiently answering my questions. I would like to thank Suvrat Raju, Shubho Roy, Debajyoti Sarkar, and Ronak Soni for several illuminating discussions. I would also like to thank the two anonymous referees for correcting several errors and pointing out a number of typos in the original draft.

These lectures were delivered at the ST4 meeting, 2018 held in NISER Bhubaneshwar. ST4 is a great platform for budding researchers to learn from each other. Many thanks to all the organizers and participants of ST4 2018 and a big thank you to everyone involved in making ST4 happen. l am grateful to the faculty and administration of NISER Bhubaneshwar for hospitality during the workshop ST4 2018, where these lectures were delivered.

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
