# Peer review of "Lectures on Bulk Reconstruction"

_SciPost Physics Lecture Notes, doi:SciPost Phys. Lect. Notes 22 (2021)_

## Round 3 · Referee Report · Anonymous (Referee 1) · 2020-6-11

Strengths

1- some nice basic points that are often assumed in the literature are explained 2- calculations are succinct and to the point

Weaknesses

1- refereed version lacks polish; typos, many careless formatting and presentation mistakes, quantities not defined, etc.

Report

These lecture notes give a brief introduction to several topics in bulk reconstruction, that is, the theory of how one represents bulk gravitational quantities in terms of dual conformal field theoretic quantities within the AdS/CFT correspondence. The notes are mostly concerned with reconstructing bulk scalar fields in terms of boundary CFT operators, covering the original HKLL prescription, as well as some of its extensions (e.g. interacting scalar fields) and other topics (e.g. symmetry-based methods). The notes offer a succinct account of the subject with several subtleties clearly exposed, and they should be accessible to anyone who has a basic understanding of the AdS/CFT correspondence and a solid background in relativity, quantum field theory in curved space-time, and CFT basics.

An overall criticism that I have of these notes is that they lack polish, to the point that they are not suitable for publication in their current form. The text contains numerous small issues such as: typos in both the text and equations, grammar and spelling issues, incorrectly-formatted punctuation, inconsistent italicization of variables and symbols, awkward equation formatting, inconsistent paragraph indentation, labels in figures inconsistent with the text, etc. Therefore, I would request that the author carefully go through the manuscript to fix these presentation issues before the next round of refereeing, at which point I would be happy to help identify remaining issues.

The content and scope of the review is appropriate, although I would recommend that the author say a few more words about entanglement wedge reconstruction besides the last three sentences at the bottom of p. 17. At the very least, the author should explain what the entanglement wedge is, that proofs are based on techniques from quantum error correction applied to holography, and give references (suggestion: 1411.7041, 1601.05416). This can be done without going into too much detail. Otherwise, while the notes mostly focus on reconstructing bulk scalar fields, the author should also mention that other types of reconstruction have been studied, such as reconstruction of fields with spin (1204.0126, 1912.00952) and reconstruction of the bulk metric (1605.01070, 1801.07280, 1904.04834).

Requested changes

  1. Abstract, Preamble 0.1- Abstract: The second sentence is a bit misleading; we don't know how to ask all bulk questions in the boundary, but we certainly know how to ask some bulk questions in the boundary. 0.2- p.2 parag.1: I do not agree; for example, studying quantum aspects of black holes using tools from the CFT has been an important part of AdS/CFT since its earliest days. Better to simply say that the duality goes both ways and leave it at that. 0.3- p.2 list: Citations needed for AMPS, black hole information problem.

  2. Overview of the the program 1.1- Typo in section title (the the). 1.2- p.3 eq. 1 and eq. 3: Do you really mean the limit as $r_1, r_2, \dots r_n \rightarrow \infty$? This should be explained and the equations adjusted accordingly. The coordinates should also be explained. 1.3- p.3 below eq. 1: Specify that $d$ is the number of space dimensions in the bulk. 1.4- p.3 last unnumbered equation: TFD is not normalized, meaning of $E$ should be explained. 1.5- p.3: A reference and/or substantiation should be given for the claim "A generic CFT state will be dual to a black hole geometry." More accurately, a generic state within a certain energy window will be dual to a black hole of the corresponding mass. 1.6- p.4: $\ell$ should be defined when it is first introduced here. 1.7- p.4 above Eq. 4: The meaning of $N$ and relation to central charge should be explained. Perhaps also consider briefly discussing the stringy origins of the duality? 1.8- p.4 below Eq. 4: There is also the 't Hooft coupling, which controls higher-derivative terms on the gravity side. 1.9- p.5 Eqs 5-7: The $\sqrt{-g}$ is missing in the last three terms. What about the $O(1/\sqrt{G})$ term? Formatting: this should be one numbered equation instead of three. 1.10- p.5: No need to repeat Eq. 8 here. 1.11- p.5 Eq. 9: $t_2$ is duplicated

  3. Boundary representation of free fields in the bulk 2.1- p.5 above Eq. 11: In the interest of pedagogy, perhaps consider explaining what is operationally meant by "taking the $G \rightarrow 0$ limit". 2.2- p.5 Eq. 11: Define $\Box$. 2.3- p.6 above Eq. 12: give ranges of $l, \vec{m}$, and later the range of $n$. 2.4- p.6 Eq. 15: Exponent should be $\Delta - d$ 2.5- p.6 Eq. 19: $t_2$ again duplicated. 2.6- p.8 Eq. 28: Second instance of $g_{\omega l \vec{m}}$ should not be conjugated. 2.7- p.8 Eq. 31: Missing a factor of 2 relative to Eq. 29. 2.8- p.8 below Eq. 31: Give citations for HKLL. 2.9- p.9, parag.1: Could you elaborate on the statement that "No modes between $-\Delta$ and $\Delta$ appear..."? This isn't clear from any of the mode expansions given. 2.10- p.9, parag.2: You should either explain how spacelike separated support can be arranged, or point to the appropriate section of the Harlow notes for details.

  4. Boundary representation for interacting fields 3.1- Correct terminology is "Green's function" 3.2- p.11 Eq. 37: Second line should read "$G(y,y') = 0$ for..." 3.3- p.11 equation below (37): Define $n^\mu$, $m$ should be $M$? 3.4- p.11 Eq. 39: What is $\Delta_-$? 3.5- p.12 Eq. 41: $m$ appears again.

  5. Reconstruction of interacting gauge and gravitational fields 4.1- p.13 parag.1: Write down the action for $\phi$ and $A_\mu$. 4.2- p.13 Eq. 48: Define the coordinates. 4.3- p.14 Eq. 49: Define $g(x,x')$. Should the coupling constant underneath be $q$ instead of $g$?

  6. Reconstruction in AdS-Rindler and causal wedge reconstruction conjecture 5.1- p.15 Eq. 50: RHS should be $-1$. 5.2- p.15 below Eq. 50: Specify that it's a uniformly accelerated observer. Give the physical meaning of $\xi$ and $\tau$. 5.3- p.15 Eq. 55: Should be $- \xi^2 d\tau^2$, the round metric in brackets should be in terms of $\chi$ instead of $\xi$. 5.4- p.15 under Eq. 55: Refer back to where $(r,t,\theta)$ are defined. 5.5- p.17 Eq. 58: $t$ should actually be $t'$ in the exponents. 5.6- Section 5.2: Highlight the difference between what is proof and what is conjecture. What specifically about the proposal is conjectural? (E.g. that reconstruction is possible for other backgrounds besides pure AdS.) 5.7- p.17 under Figure 5: define $X$ and $Y$ in the text, not just implicitly from the figure. 5.8- See general comment about entanglement wedge reconstruction.

  7. Scalar field reconstruction from symmetries 6.1- p.18 Eq. 59: cosh is missing an argument 6.2- p.18 Eqs. 60,61: Perhaps best to omit index on the CFT generator. 6.3- p.19 parag.1: Define $K$, $P$ and $D$, and later $M$. 6.4- p.19 first unnumbered equation: This equation needs a careful clean-up. 6.5- p.19 second unnumbered equation: $x$ shouldn't carry an index; there is only one such coordinate in AdS2. 6.6- p.19 Eq. 64: spacing needed 6.7- p.19 Eq. 67: $\Delta_\mathcal{O}$ has not been defined. 6.8- p.19 Eq. 68: What index is being summed over? 6.9- p.20 Eq. 69: On the LHS should it be $x_a$? 6.10- p.20 under Eq. 70: $d$ is the number of bulk space dimensions. 6.11- p.20 Eq. (71): It should be $\phi_{CFT}$, on RHS $\Delta_\phi - d$ in brackets. 6.12- p.20 Eq. 72: $\Delta_\mathcal{O}$ appears again, still undefined. 6.13- p.21 under Eq. 75: instead of ``the numbers'', say that $a_n$ matches (67).

  8. Challenges to bulk reconstruction 7.1- p.21 first parag. in section: so far we have worked in the formally infinite $N$ regime, not just the large $N$ regime. 7.2- p.21 start of 7.1: (large) spatial volumes in $AdS_{d+1}$ do not scale like $L^d$; they scale like $L^{d-1}$, so this argument is incorrect. E.g. take a constant-$t$ slice in the metric (10) and consider a sphere of radius $L$ centred at $r=0$. Its volume goes like \begin{equation} \int_0^L dr~ \frac{r^{d-1}}{\sqrt{1+r^2}} \sim L^{d-1} ~ \text{for large} ~ L \end{equation} 7.3- top of p.22: Define $E$, and include appropriate factors of $G$ so that it can be compared to $S_{BH}$ below. 7.4- p.23 Fig 7: Label the quadrants $I$ through $IV$. 7.5- p.24 last parag.: It seems more natural to first define pure black hole microstates and then say that the black hole approaches an equilibrium configuration of these microstates on intermediate time-scales rather than the other way around. Likewise, the metric only takes this approximate time-independent form on intermediate time-scales: long enough for the black hole to equilibrate, but not long enough for it to significantly evaporate. 7.6- p.25 Eq 82, 83: The sum over $l$ and $\vec m$ is missing. What is the origin of the phase $\delta_\omega$? 7.7- p.25 last parag.: How are you discretizing the modes? What difference does choosing one discretization method over another make? 7.8- p.26 Eq. 88: Define $k$. 7.9- p.26 below Eq. 88: what does "small" mean in "small powers of the CFT Hamiltonian"? 7.10- p.26 below Eq. 88: "The upshot of these two conditions..." the in-line math expressions in this sentence need some punctuation and/or words. 7.11- p.26 Eq. 89: $A$ hasn't been defined, span takes as its argument a set. 7.12- p.26 below Eq. 89: If $H_\psi$ is not a Hilbert space, how is its dimension defined? 7.13- p.26 parag. below Eq. 90: This paragraph about solving systems of equations is unclear. One way to clarify could be to write out in equations what is being described. 7.14- p.27 Eq. 91: Rightmost expression is incorrect (the rightmost $b$ should be $b_\omega$, I believe). 7.15- p.28 Eq. 95: TFD is not normalized, exponent should be $-\beta E/2$. 7.16- p.28 below Eq. 95: A factorized state can only be interpreted as two disconnected geometries provided there exist semiclassical bulk duals. Provide a reference for $|E\rangle|E\rangle$ being dual to geometries.

  • validity: good
  • significance: -
  • originality: -
  • clarity: good
  • formatting: below threshold
  • grammar: reasonable

Author:  Nirmalya Kajuri  on 2020-07-15  [id 887]

(in reply to Report 1 on 2020-06-11)

We would like to thank the reviewer for the very detailed and very helpful review. We have attached the list of changes with this response.

Our response to the referee's points is as follows:

0.1- Abstract: The second sentence is a bit misleading; we don't know how to ask all bulk questions in the boundary, but we certainly know how to ask some bulk questions in the boundary.

Response: Corrected to "The CFT may well have all the answers, but we don't know how to ask all the right questions!"

0.2- p.2 parag.1: I do not agree; for example, studying quantum aspects of black holes using tools from the CFT has been an important part of AdS/CFT since its earliest days. Better to simply say that the duality goes both ways and leave it at that.

Response: Corrected. That statement has been removed.

0.3- p.2 list: Citations needed for AMPS, black hole information problem.

Response: Citations added.

  1. Overview of the the program 1.1- Typo in section title (the the).

R: Typo corrected.

1.2- p.3 eq. 1 and eq. 3: Do you really mean the limit as r_1,r_2...

R: Corrected. All the radial distances are r, and r goes to infinity.

1.3- p.3 below eq. 1: Specify that d is the number of space dimensions in the bulk.

R: Corrected.

1.4- p.3 last unnumbered equation: TFD is not normalized, meaning of E should be explained.

Normalization term included, clarified that |E> are energy eigenstates.

1.5- p.3: A reference and/or substantiation should be given for the claim "A generic CFT state will be dual to a black hole geometry." More accurately, a generic state within a certain energy window will be dual to a black hole of the corresponding mass.

This statement is removed. 1.6- p.4: ℓ should be defined when it is first introduced here.

Corrected, ℓ introduced at this place.

1.7- p.4 above Eq. 4: The meaning of N and relation to central charge should be explained. Perhaps also consider briefly discussing the stringy origins of the duality?

This was already mentioned, but we have clarified further.

1.8- p.4 below Eq. 4: There is also the 't Hooft coupling, which controls higher-derivative terms on the gravity side.

1.9- p.5 Eqs 5-7: The √g is missing in the last three terms. What about the O(1/√G) term? Formatting: this should be one numbered equation instead of three.

Corrected. The O(1/√G) term does not appear beccause we have not expanded the gravitational action.

1.10- p.5: No need to repeat Eq. 8 here.

Corrected.

1.11- p.5 Eq. 9: t2 is duplicated

Corrected.

  1. Boundary representation of free fields in the bulk 2.1- p.5 above Eq. 11: In the interest of pedagogy, perhaps consider explaining what is operationally meant by "taking the G →0 limit".

We have added that "In this limit gravity is switched off. So we can neglect gravity and consider the scalar field in a fixed background."

2.2- p.5 Eq. 11: Define box operator.

We have added that "where $\Box$ is the D'Alembartian in anti-de Sitter spacetime."

2.3- p.6 above Eq. 12: give ranges of l...

Ranges added.

2.4- p.6 Eq. 15: Exponent should be Δ − d

Corrected.

2.5- 2.7 : corrected.

2.8- p.8 below Eq. 31: Give citations for HKLL.

Citations added.

2.9- p.9, parag.1: Could you elaborate on the statement that "No modes between −Δ and Δ appear..."? This isn't clear from any of the mode expansions given.

We have rewritten this for clarification: We note that the smearing function is not unique. We can see from \eqref{molution} that modes between $-\Delta$ and $\Delta$ appear in the solution for $\mathcal{O}(t,\Omega)$. Therefore if we add any $e^{iJ t}$ to the solution where $J$ is an integer between $-\Delta + 1$ and $\Delta - 1$, the integration of its product with $\mathcal{O}(r,t,\Omega)$ vanishes.

2.10- p.9, parag.2: You should either explain how spacelike separated support can be arranged, or point to the appropriate section of the Harlow notes for details.

We have added a reference to the appropriate section of the original HKLL paper.

  1. Boundary representation for interacting fields 3.1- Correct terminology is "Green's function"

Corrected. 3.2- 3.3 3.4- p.11 Eq. 39: What is Delta_?

Replaced by (d -\Delta).

3.5- p.12 Eq. 41: m appears again.

Corrected.

  1. Reconstruction of interacting gauge and gravitational fields 4.1- p.13 parag.1: Write down the action for ϕ and A_μ.

Action added. 4.2- p.13 Eq. 48: Define the coordinates.

Rewritten in a coordinate invariant form. 4.3- p.14 Eq. 49: Define g(x,x′). Should the coupling constant underneath be q instead of g?

It has been clarified that g(x,x') is any condition satisfying the equation below equation (48).

  1. Reconstruction in AdS-Rindler and causal wedge reconstruction conjecture 5.1- p.15 Eq. 50: RHS should be −1.

Corrected.

5.2- p.15 below Eq. 50: Specify that it's a uniformly accelerated observer. Give the physical meaning of and τ.

Corrected. Physical meanings of rindler coordinates in terms of acceleration and proper time of rindler observer given. . 5.3- 5.5: Corrected.

5.6- Section 5.2: Highlight the difference between what is proof and what is conjecture. What specifically about the proposal is conjectural? (E.g. that reconstruction is possible for other backgrounds besides pure AdS.)

We have clarified this. The relevant section now reads:

The AdS-Rindler reconstruction can be extended to a more general class of bulk regions -- the causal wedges of ball-shaped boundary regions ( For AdS$_3$ these are just intervals on the boundary). An AdS/Rindler chart can be defined on such wedges \cite{Casini:2011kv,Morrison:2014jha} and the above method applied. This result leads to the causal wedge reconstruction conjecture.

The causal wedge reconstruction conjecture holds that any field at any point in the causal wedge of any boundary region R in an asymptotically AdS spacetime can be reconstructed as an operator in the boundary region $D[R]$. The intuitive explanation for this is that any point in $C[R]$ can be accessed by a causal observer starting from and returning to $D[R]$. But as the boundary theory is unitary, it already knows the information such an observer may bring. Thus the information about the entire causal wedge is already present in R. As noted, it is proved only for causal wedges of ball-shaped subregions in pure Anti-de Sitter spacetimes. For more general wedges in AdS, as well as for any causal wedges in more general asymptotically AdS spacetimes, this remains a conjecture.
For more general wedges in AdS, as well as for any causal wedges in more general asymptotically AdS spacetimes, this remains a conjecture.

5.7- p.17 under Figure 5: define X and Y in the text, not just implicitly from the figure.

This has been done.

5.8- See general comment about entanglement wedge reconstruction.

We have added a few lines on entanglement wedge reconstruction at the end of the section on causal wedge reconstruction. 6. Scalar field reconstruction from symmetries 6.1-6.7 : Typos corrected. 6.7- This was a typo for \Delta_\phi which has been corrected. 6.8- p.19 Eq. 68: What index is being summed over?

Index added.

6.9- p.20 Eq. 69: On the LHS should it be x_a?

Yes, corrected.

6.10- 6.13: corrections made.

  1. Challenges to bulk reconstruction 7.1- p.21 first parag. in section: so far we have worked in the formally infinite N regime, not just the large N regime.

Corrected. 7.2- This argument has been removed.

7.3- top of p.22: Define E, and include appropriate factors of G so that it can be compared to SBH below.

This has been done. 7.4- p.23 Fig 7: Label the quadrants.

Labelled. . 7.5- p.24 last parag.: It seems more natural to first define pure black hole microstates and then say that the black hole approaches an equilibrium configuration of these microstates on intermediate time-scales rather than the other way around. Likewise, the metric only takes this approximate time-independent form on intermediate time-scales: long enough for the black hole to equilibrate, but not long enough for it to significantly evaporate.

Response: We are considering big black holes which do not evaporate. We have added the following definition of equilibrium states: "Here we introduce the notion of 'equilibrium states' -- states which are dual to black holes which are in approximate equiilibrium. The defining property of equilibrium states is that the elements of the small algebra will have approximately (i.e up to $1/N$ corrections) thermal correlators for these states. Note that there can be more than one equilibrium state dual to a given black hole geometry."

7.6- p.25 Eq 82, 83: The sum over l and ms missing. What is the origin of the phase δω?

Sum added. The origin of the phase is the normalizability criterion, only a linear combination of the two modes is normalizable at the boundary, and that introduces the phase factor. This has been explained in the text.

7.7- p.25 last parag.: How are you discretizing the modes? What difference does choosing one discretization method over another make?

The discretization has been explained in the text.

7.8- p.26 Eq. 88: Define k.

This has been done.

7.9- p.26 below Eq. 88: what does "small" mean in "small powers of the CFT Hamiltonian"?

small means much less than the central charge, we have added this.

7.10- p.26 below Eq. 88: "The upshot of these two conditions..." the in-line math expressions in this sentence need some punctuation and/or words.

Corrected. 7.11- p.26 Eq. 89: A hasn't been defined, span takes as its argument a set.

This was a typo, corrected. 7.12- p.26 below Eq. 89: If H ψ is not a Hilbert space, how is its dimension defined?

We had incorrectly written that it is not a Hilbert space, this has been corrected.

7.13- p.26 parag. below Eq. 90: This paragraph about solving systems of equations is unclear. One way to clarify could be to write out in equations what is being described.

We have rewritten this aprt as follows:

How do we know that mirror operators will exist? In other words, how doe we know that one can always find $\mb_\omega$ and $ \mb^\dagger_\omega$ that satisfy \eqref{mirror}? As all $B_\alpha |\psi\rangle$ are linearly independent, \eqref{mirror} defines the operators $\mb_\omega$ and $ \mb^\dagger_\omega$ by a set $D_B$ equations each. But the dimension of the full Hilbert space is approximately $e^{N^2}$ so the operators $\mb_\omega$ and $ \mb^\dagger_\omega$ can be thought of as a $e^{N^2} X e^{N^2}$ matrices. But as $D_B \ll e^{N^2}$, solutions can always be found for the above set of equations.

7.14- p.27 Eq. 91: Rightmost expression is incorrect (the rightmost

Corrected.

7.15- p.28 Eq. 95: TFD is not normalized, exponent should be ...

Corrected.

7.16- p.28 below Eq. 95: A factorized state can only be interpreted as two disconnected geometries provided there exist semiclassical bulk duals.

The assumption in the original paper was that geometries dual to the eigenstates can be quantum. We have clarified this.

Attachment:

list_H4O1LgR.pdf

---

## Round 3 · Referee Report · Anonymous (Referee 2) · 2020-6-17

Strengths

  1. An accessible and well-organized account of results that are scattered across a large and not very user-friendly literature.

Weaknesses

  1. Some approaches to bulk reconstruction receive little or no mention, so it should be understood that these notes emphasize a coherent point of view rather than an exhaustive treatment of the subject.

Report

These notes, based on lectures at the ST4 2018 meeting, provide a valuable service to the community by collecting and summarizing results on bulk reconstruction that are scattered across a large literature.

Bulk reconstruction is a large subject, very much a work in progress, with a diversity of approaches being pursued. The treatment here is somewhat selective. It's a bit unfortunate that certain approaches (such as entanglement wedge reconstruction and the role of quantum error correction, or OPE blocks and kinematic space) receive little or no mention.

But as a positive, this selective approach makes for a coherent and accessible presentation. The notes begin with a statement of the AdS/CFT extrapolate dictionary then cover global reconstruction for free scalar fields. Interacting scalar fields are disussed to ${\cal O}(1/N)$, from the point of view of both bulk equations of motion and microcausality. The issues affecting fields with bulk gauge redundancy are discussed, causal wedge reconstruction is outlined, and the use of little group symmetries to build bulk fields is presented. The last section presents some open questions and issues in bulk reconstruction: finite $N$ effects, the black hole interior (with an outline of the Papadodimas - Raju proposal), and the Marolf-Wall paradox.

The discussion tends to outline a topic and leave details to the references. So these notes will be particularly useful for readers who are either new to the field or looking for an entry to the literature.

Requested changes

There are a few typos and misconceptions I think should be corrected.

p. 7: "lives in" -> "acts on"

p. 10: The statement that Poincare coordinates give the same result isn't quite accurate, since Poincare coordinates lead to a smearing function with support on the boundary of the Poincare patch. The different represenations can be related with the help of the antipodal map and the sort of non-uniqueness mentioned on p. 9. See for example appendix C of ref. 15.

bottom p. 12: it doesn't affect the 2-point function, but more importantly it doesn't affect the extrapolate dictionary since $\phi^{(1)}$ dies off faster than $\phi^{(0)}$ near the boundary

p. 13, last sentence of section 3: at least in principle higher point correlators are determined by the OPE. So in this sense the microcausality approach could be used to determine higher-order corrections to bulk observables.

p. 14, last full paragraph: Einstein's equations coupled to matter fields have a gauge symmetry of diffeomorphism invariance. But I don't think one can reverse the logic - diffeomorphism invariance doesn't require a dynamical metric. So I find it confusing to say that we have diffeomorphism invariance only when gravity is dynamical and not when we do field theory in a given background.

p. 20: Belstrami -> Beltrami. Also, bottom of the page: a similar calculation appeared in hep-th/1604.07383, equation (2.5).

p. 21: it's worth noting that refs. [37,38] were able to take gravitational dressing into account

p. 21: the comparison of bulk and boundary degrees of freedom is a bit subtle in AdS, since area and volume scale the same way. See for example the top of p. 8 in hep-th/9805114.

p. 22: I think there's a typo in (77): the last $G^d$ should be $G^{1/d}$. Also it should be clarified that the relation $r_s^d \approx G M$ below (77) is an approximation, valid for AdS-Schwarzschild black holes in the limit of large Schwarzschild radius.

p. 29: near the boundary -> near the singularity

  • validity: -
  • significance: -
  • originality: -
  • clarity: -
  • formatting: -
  • grammar: -

Author:  Nirmalya Kajuri  on 2020-07-15  [id 888]

(in reply to Report 2 on 2020-06-17)

We thank the referee for their helpful report. We have attached the full list of changes here. We address their points below:

p. 7: "lives in" -> "acts on".
Response: Corrected.

p. 10: The statement that Poincare coordinates give the same result isn't quite accurate, since Poincare coordinates lead to a smearing function with support on the boundary of the Poincare patch. The different represenations can be related with the help of the antipodal map and the sort of non-uniqueness mentioned on p. 9. See for example appendix C of ref. 15.
Response: Corrected.

We have changed the paragraph to

"Here we worked in global coordinates but we could have worked in the Poincare coordinates. That gives a smearing function with support on the boundary of the Poincare patch. This matches with the global smearing function in the Poincare patch coordinates, up to the ambiguities in the defintion of smearing function mentioned above. We refer to appendix C of [18]."

Here [18] is the reference mentioned by there referee.

bottom p. 12: it doesn't affect the 2-point function, but more importantly it doesn't affect the extrapolate dictionary since ...

Response: corrected. We have changed the paragraph to:

"Where $\mathcal{O}_{\Delta_n}(X')$ are higher dimensional primaries. This is a natural guess, because $\phi^{1}(y)$ falls off faster than $\phi^{(0)}(y)$ near the boundary and does not affect the extrapolate dictionary. Further, it doesn't affect the two point function (primaries of different dimensions have vanishing two point functions)."

p. 13, last sentence of section 3: at least in principle higher point correlators are determined by the OPE. So in this sense the microcausality approach could be used to determine higher-order corrections to bulk observables.

Response: Here our point was that one needs the details of the theory to determine four-point functions, only causality won't do. Perhaps the author is referring to the paper by Kabat et al on 'bulk OPE's. We believe this paper to be incorrect, and one of the authors agreed with our criticism (private communication).

p. 14, last full paragraph: Einstein's equations coupled to matter fields have a gauge symmetry of diffeomorphism invariance. But I don't think one can reverse the logic - diffeomorphism invariance doesn't require a dynamical metric. So I find it confusing to say that we have diffeomorphism invariance only when gravity is dynamical and not when we do field theory in a given background.

Response: Here our point was that field theory in curved spacetime is not diffeo invariant. We have added a footnote to clarify this:

"One can obtain a diffeomorphism-invariant formulation of a field theory in a given background by introducing auxillary variables. This gives us parametrized field theories. The above comment is about when we do field theory without introducing such auxillary variables."

p. 20: Belstrami -> Beltrami. Also, bottom of the page: a similar calculation appeared in hep-th/1604.07383, equation (2.5).

Response: Corrected, reference added.

p. 21: it's worth noting that refs. [37,38] were able to take gravitational dressing into account.

Response: Yes, but here we are referring to finite N ccorrections.

p. 21: the comparison of bulk and boundary degrees of freedom is a bit subtle in AdS, since area and volume scale the same way. See for example the top of p. 8 in hep-th/9805114.

Response: This paragraph has been removed.

p. 22: I think there's a typo in (77): ..... below (77) is an approximation, valid for AdS-Schwarzschild black holes in the limit of large Schwarzschild radius.

Response: Typo corrected, the point about approximation clarified.

p. 29: near the boundary -> near the singularity

Response: Corrected.

Attachment:

list_Zcfjs1V.pdf

---

## Round 4 · Referee Report · Anonymous (Referee 2) · 2020-7-25

Report
This article provides a well-written, accessible and coherent perspective on bulk reconstruction. The strengths, weaknesses and my overall summary remain as before. But the current version has been corrected and improved in many small ways. It will be particularly useful for readers who are new to the field or are looking for an entry to the literature.

Anonymous on 2020-07-15 [id 886]
Hi
I am attaching the list of changes here.
Attachment:
list.pdf

---

## Round 4 · Referee Report · Anonymous (Referee 1) · 2020-7-30

Report
This is my second round of refereeing of this manuscript (reflecting arXiv v4). While I am largely satisfied with the author's responses and corrections, there are still some lingering issues.
p.5 Eq. 6: If the gravitational action has not been expanded, then the question remains why there isn't a $O(N)$ term, as one would expect from a generic expansion in $N$. If the answer is that there cannot be such a term so that the CFT is in correspondence with an action that describes pure Einstein gravity plus a scalar field, then is this an assumption? Can you check this in examples? The example that I'm aware of is the partition function of 2D QCD (hep-th/9301068), but I would ask that the author comment further on this consideration.
p.9 below Eq. 30: I think it should read "We can see from (14) that modes between $-\Delta$ and $\Delta$ do not appear..."
p.9 above Eq. 31: Open bracket, incorrect capitalization, "too"
p. 18 penultimate parag.: incorrect spacing, capitalization in parentheses
p.20 above Eq. 62: \times instead of $X$, should read $\mathbb{R}$ instead of $R$ throughout paragraph.
p.26 footnote 8: "rRemember"
p. 28 below Eq. 90: My comment 7.10 has not been satisfactorily addressed. The sentence still contains typos and grammar issues.
p.28 parag. above Eq. 91: "equiilibrium"
p. 29 below Eq. 92: "how doe", again use \times instead of $X$
I again urge the author to carefully go through the manuscript and fix presentation issues, as I requested in my original report. In particular, the author should pay close attention to typos, punctuation, italicization, and indentation.
p.5 Eq. 6: If the gravitational action has not been expanded, then the question remains why there isn't a $O(N)$ term, as one would expect from a generic expansion in $N$. If the answer is that there cannot be such a term so that the CFT is in correspondence with an action that describes pure Einstein gravity plus a scalar field, then is this an assumption? Can you check this in examples? The example that I'm aware of is the partition function of 2D QCD (hep-th/9301068), but I would ask that the author comment further on this consideration.
p.9 below Eq. 30: I think it should read "We can see from (14) that modes between $-\Delta$ and $\Delta$ do not appear..."
p.9 above Eq. 31: Open bracket, incorrect capitalization, "too"
p. 18 penultimate parag.: incorrect spacing, capitalization in parentheses
p.20 above Eq. 62: \times instead of $X$, should read $\mathbb{R}$ instead of $R$ throughout paragraph.
p.26 footnote 8: "rRemember"
p. 28 below Eq. 90: My comment 7.10 has not been satisfactorily addressed. The sentence still contains typos and grammar issues.
p.28 parag. above Eq. 91: "equiilibrium"
p. 29 below Eq. 92: "how doe", again use \times instead of $X$
I again urge the author to carefully go through the manuscript and fix presentation issues, as I requested in my original report. In particular, the author should pay close attention to typos, punctuation, italicization, and indentation.
Requested changes
1- Itemized comments in report 2- Fix typos, presentation, and formatting issues throughout

---

## Round 7 · Referee Report · Anonymous · 2021-1-4

Report

This is my third round of refereeing of this manuscript (reflecting arXiv v7).
I am satisfied with the author's replies and I do not think it is necessary for me to pursue further questions. Two typos that were introduced on p.5 are as follows:

p.5 parag. 3: "Note that we have N is the only expansion parameter so far in the CFT." (Delete "we have").

p.5: $N$ is inconsistently italicized on this page and elsewhere.

At this point, I will leave further formatting issues to the editorial and typesetting teams' discretion.

---

## Round 7 · Referee Report · Anonymous · 2021-1-14

Report

I concur with the first referee and am satisfied with the author's replies. In the refereeing process several conceptual points have been clarified and numerous typos have been corrected. I recommend proceeding with publication of the current version.

---

## Round 7 · Author Response

I have corrected lingering typos and grammatical mistakes including the ones pointed out by the referee. To address the referee's comment we have added the following paragraph:

"For a CFT to have a semiclassical bulk dual, a set of conditions have to be fulfilled. We
refer the reader to [15, 16] for discussions on this issue. One key condition is that the CFT
should have a parameter N  1 which controls the factorization of the correlators of the
primary operators which are dual to bulk fields. This means if the two-point function of such
a primary operator is normalized to be of O(1), then higher correlators are suppressed as
hOiOjOki ∼ 1/Na
where a is some O(1) number. In particular, the correlators will follow Wick contraction for
infinite N. N is dual to the perturbative parameter in the bulk field theory, where a similar
factorization takes place in bulk correlators."

We hope this addresses the referee's question. The point is that if two-point functions in the CFT are normalized to order one, then higher order correlators are suppressed by powers of N. The corresponding object in the bulk dual theory is the coupling constant which plays an equivalent role in the factorization of bulk field correlators. We refer the referee to the discussion in section 2.4 of Daniel Harlow's TASI lectures. If this explanation is not satisfactory we can remove the action term from our notes.

---

## Round 7 · List of Changes

1. We have corrected typos and grammatical mistakes that were present in the last manuscript, including the ones pointed out by the referee.

2. In response to the referee's first point, we have added to the first paragraph in page 5 which now reads:
"For a CFT to have a semiclassical bulk dual, a set of conditions have to be fulfilled. We
refer the reader to [15, 16] for discussions on this issue. One key condition is that the CFT
should have a parameter N  1 which controls the factorization of the correlators of the
primary operators which are dual to bulk fields. This means if the two-point function of such
a primary operator is normalized to be of O(1), then higher correlators are suppressed as
hOiOjOki ∼ 1/Na
where a is some O(1) number. In particular, the correlators will follow Wick contraction for
infinite N. N is dual to the perturbative parameter in the bulk field theory, where a similar
factorization takes place in bulk correlators."

---

## Editorial Decision

published